METHODS AND RESOURCES

# A proteomic atlas of senescence-associated secretomes for aging biomarker development

Nathan Basisty[1], Abhijit Kale[1], Ok Hee Jeon[1¤], Chisaka Kuehnemann[1], Therese Payne[1], Chirag Rao[1], Anja Holtz[1], Samah Shah[1], Vagisha Sharma[2], Luigi Ferrucci[3], Judith Campisi[1,4], Birgit Schilling[1]*

1 The Buck Institute for Research on Aging, Novato, California, United States of America, 2 University of Washington, Seattle, Washington, United States of America, 3 Intramural Research Program of the National Institute on Aging, NIH, Baltimore, Maryland, United States of America, 4 Lawrence Berkeley Laboratory, University of California, Berkeley, California, United States of America

¤ Current address: Korea University College of Medicine, Seoul, Republic of Korea
* bschilling@buckinstitute.org

**Data Availability Statement:** All raw files are uploaded to the Center for Computational Mass Spectrometry, MassIVE, and the ProteomeXchange Consortium and can be

## Abstract

The senescence-associated secretory phenotype (SASP) has recently emerged as a driver of and promising therapeutic target for multiple age-related conditions, ranging from neuro-degeneration to cancer. The complexity of the SASP, typically assessed by a few dozen secreted proteins, has been greatly underestimated, and a small set of factors cannot explain the diverse phenotypes it produces in vivo. Here, we present the "SASP Atlas," a comprehensive proteomic database of soluble proteins and exosomal cargo SASP factors originating from multiple senescence inducers and cell types. Each profile consists of hundreds of largely distinct proteins but also includes a subset of proteins elevated in all SASPs. Our analyses identify several candidate biomarkers of cellular senescence that overlap with aging markers in human plasma, including Growth/differentiation factor 15 (GDF15), stanniocalcin 1 (STC1), and serine protease inhibitors (SERPINs), which significantly correlated with age in plasma from a human cohort, the Baltimore Longitudinal Study of Aging (BLSA). Our findings will facilitate the identification of proteins characteristic of senescence-associated phenotypes and catalog potential senescence biomarkers to assess the burden, originating stimulus, and tissue of origin of senescent cells in vivo.

## Introduction

Cellular senescence is a complex stress response that causes an essentially irreversible arrest of cell proliferation and development of a multicomponent senescence-associated secretory phenotype (SASP) [1–4]. The SASP consists of a myriad of cytokines, chemokines (CXCLs), growth factors, and proteases that initiate inflammation, wound healing, and growth responses in nearby cells [5,6]. In young healthy tissues, the SASP is typically transient and tends to contribute to the preservation or restoration of tissue homeostasis [5]. However, senescent cells increase with age, and a chronic SASP is known or suspected to be a key driver of many pathological hallmarks of aging, including chronic inflammation, tumorigenesis, and impaired stem

downloaded using the following links: ftp://massive.ucsd.edu/MSV000083750 and http://proteomecentral.proteomexchange.org/cgi/GetDataset?ID=PXD013721 (MassIVE ID number: MSV000083750; ProteomeXchange ID number: PXD013721). Data uploads include the protein identification and quantification details, spectral library, and FASTA file used for mass spectrometric analysis. SASP proteomic profiles are available on Panorama (https://panoramaweb.org/project/Schilling/SASP_Atlas_Buck/begin.view?), a repository for targeted mass spectrometry assays generated in Skyline software. All data are available for viewing and downloading on SASP Atlas (www.saspatlas.com).

**Funding:** This work was supported by grants from the National Institute on Aging (BS is supported by U01 AG060906-02, Principal Investigator: Schilling; JC is supported by P01AG017242 and R01AG051729, Principal Investigator: Campisi) and a National Institutes of Health Shared Instrumentation Grant (BS is supported by 1S10 OD016281, Buck Institute). NB and OHJ were supported by postdoctoral fellowships from the Glenn Foundation for Medical Research. AK was supported by the SENS Foundation. VS was supported by the University of Washington, Seattle Proteomics Resource (UWPR95794). The funders had no role in study design, data collection and analysis, decision to publish, or preparation of the manuscript.

**Competing interests:** I have read the journal's policy and the authors of this manuscript have the following competing interests: JC is a founder and shareholder of Unity Biotechnology, which develops senolytic drugs. All other authors have declared no competing interests.

**Abbreviations:** ATV, atazanavir treatment; BLSA, Baltimore Longitudinal Study of Aging; CALR, calreticulin; CCL3, C-C motif chemokine 3; CST4, Cystatin-S; CXCL, chemokine; CXCL1, chemokine C-X-C motif ligand 1; DAMP, damage-associated molecular pattern; DDA, data-dependent acquisition; DIA, data-independent acquisition; DMEM, Dulbecco's Modified Eagle Medium; ECM, extracellular matrix; eSASP, extracellular vesicle senescence-associated secretory phenotype; EV, extracellular vesicle; FBS, fetal bovine serum; GDF15, Growth/differentiation factor 15; HMGB1, high mobility group box 1 protein; IGFBP, IGF binding protein; IL-6, interleukin 6; IR, X-irradiation; LAMB1, laminin subunit beta-1; MMP, matrix metalloproteinase; MS2, tandem mass spectrometry; qRT-PCR, quantitative real-time PCR; RAS, inducible RAS overexpression; SA-β-Gal, senescence-associated β-galactosidase; SASP,

cell renewal [5,7]. Powerful research tools have emerged to investigate the effect of senescence on aging and disease, including two transgenic p16$^{INK4a}$ mouse models that allow the selective elimination of senescent cells [8,9] and compounds that mimic the effect of these transgenes. Data from several laboratories, including our own, strongly support the idea that senescent cells and the SASP drive multiple age-related phenotypes and pathologies, including atherosclerosis [10], osteoarthritis [11], cancer metastasis, cardiac dysfunction [12,13], myeloid skewing [14,15], kidney dysfunction [16], and overall decrements in health span [17]. Recently, senescent cells were shown to secrete bioactive factors into the blood that alter hemostasis and drive blood clotting [18]. SASP factors therefore hold potential as plasma biomarkers for aging and age-related diseases that are marked by the presence of senescent cells.

To develop robust and diverse senescence and aging biomarker candidates, a comprehensive profile of the context-dependent and heterogeneous SASP is needed. Several types of stress elicit a senescence and SASP response, which in turn can drive multiple phenotypes and pathologies associated with mammalian aging. These stressors have both shared and distinct secretory components and biological pathways. For example, telomere attrition resulting from repeated cell division (replicative senescence), ionizing radiation, chromatin disruption, and activation of certain oncogenes all can cause senescence-inducing genotoxic stresses, as can genotoxic therapeutic drugs, such as certain anticancer chemotherapies [13] and therapies for HIV treatment or prevention [19]. However, while both ionizing radiation and oncogenes lead to DNA double-strand breaks, ionizing radiation uniquely produces clustered oxidative DNA lesions [20], whereas oncogene activation drives DNA hyper-replication and double-strand breaks [21]. Whether different senescence inducers produce similar or distinct SASPs is at present poorly characterized. Thus, a comprehensive characterization of SASP components is critical to understanding how senescent responses can drive diverse pathological phenotypes in vivo.

The SASP was originally characterized by antibody arrays, which are necessarily biased, to measure the secretion of a small set of pro-inflammatory cytokines, proteases and protease inhibitors, and growth factors [1,2,4,22]. Subsequently, numerous unbiased gene expression studies performed on different tissues and donors of varying ages suggest that the SASP is more complex and heterogeneous [23]; however, a recent meta-analysis of senescent cell transcriptomes confirmed the expression of a few dozen originally characterized SASP factors in multiple senescent cell types [24].

While unbiased transcriptome analyses are valuable, they do not directly assess the presence of secreted proteins. Thus, proteomic studies are needed to accurately and quantitatively identify SASP factors as they are present in the secretomes of senescent cells. Recently, a mass spectrometric study reported several SASP factors induced by genotoxic stress [25], but an in-depth, quantitative, and comparative assessment of SASPs originating from multiple stimuli and different cell types is lacking. Senescent cells also secrete bioactive exosomes [26,27] with both protein and microRNA [28] cargos. Exosomes secreted by senescent cells have been shown to have pro-tumorigenic effects [28], are associated with osteoarthritis [29], and have the ability to induce paracrine senescence [26]. A previous proteomic analysis of protein cargo from senescent extracellular vesicles (EVs) identified few known SASP factors [26], meriting further direct proteomic comparisons between EVs and soluble SASP factors.

In this study, we demonstrate that the SASP is not a single phenotype but rather is highly complex, dynamic, and dependent on the senescence inducer and cell type. Here, we also present the "SASP Atlas" (www.SASPAtlas.com), a comprehensive, curated, and expanding online database of the soluble senescence-associated secretory phenotype (sSASP) induced by various stimuli in several cell types. We also analyzed the extracellular vesicle SASP (eSASP), which is largely distinct from the sSASP. Our approach leverages an innovative data-independent mass

senescence-associated secretory phenotype; SERPIN, serine protease inhibitor; SERPINE1, plasminogen activator inhibitor 1; sSASP, soluble senescence-associated secretory phenotype; STC1, stanniocalcin 1; TIMP, tissue inhibitor of metallopeptidase; TP53, tumor protein p53; TRPS, tunable resistive pulse sensing.

spectrometry workflow to discover new SASP biomarker candidates. The SASP Atlas can help identify candidate biomarkers of aging and diseases driven by senescent cells. We also show that the SASP is enriched for protein markers of human aging and propose a panel of top SASP-based aging and senescence biomarker candidates.

## Results

### Cellular senescence extensively alters the secreted proteome

We established an efficient and streamlined proteomic workflow to discover novel SASP factors. We collected proteins secreted by senescent and quiescent/control primary human lung fibroblasts (IMR-90) and renal cortical epithelial cells (**Fig 1**). Briefly, we induced senescence in the cultured cells by X-irradiation (IR), inducible RAS overexpression (RAS), or atazanavir treatment (ATV; a protease inhibitor used in HIV treatment) and allowed 1 to 2 weeks for the senescent phenotype to develop, as described [2]. In parallel, control cells were made quiescent by incubation in 0.2% serum for 3 days and were either mock-irradiated or vehicle treated. Treated and control cells were subsequently cultured in serum-free medium for 24 hours and the conditioned media, containing soluble proteins and exosomes/EVs, was collected. Soluble proteins and exosomes/EVs were separated by ultracentrifugation.

This label-free data-independent acquisition (DIA) approach enabled sensitive and accurate quantification of SASP proteins by integrating the tandem mass spectrometry (MS2) fragment ion chromatograms [30,31]. We quantitatively compared proteins secreted by senescent cells with controls, and significantly changed proteins (q-value <0.05) that had a fold change of at least 1.5-fold (SEN/CTL) were identified. Proteins secreted at significantly higher levels by senescent relative to quiescent cells were defined as SASP factors. In fact, most proteins were secreted at much higher levels by senescent cells compared with non-senescent cells (**Fig 2**). Each treatment and control group contained 4–10 biological replicates (see Materials and methods for replicate details and experimental design). Relative protein quantification and statistical details are presented in **S1 Table**. Induction of senescence was verified by senescence-associated β-galactosidase (SA-β-Gal) activity and p16$^{INK4a}$ and interleukin-6 (IL-6) mRNA levels (**S1A, S1B** and **S1C Fig**), as described [2]. There was no detectable difference in cell death between senescent and non-senescent cells, as measured by a Sytox Green viability dye assay (**S2 Fig**). IR and RAS overexpression induced senescence in >90% of cells and ATV induced senescence in about 65% of cells (**S1A and S1B Fig**).

This unbiased proteomic profiling identified between 441 and 1,693 secreted proteins per senescence inducer, a large fraction of which were significantly up- or down-regulated in the secretome following induction of senescence by IR, RAS, or ATV (**Fig 2**). Between 340 and 714 proteins changed significantly in response to each inducer. As expected, most of the significantly changed proteins were markedly up-regulated in the SASP from senescent compared with quiescent cells, but, interestingly, a minority were down-regulated (**Fig 2A**). Notably, the protein cargo of exosomes/EVs released by senescent cells was distinct compared with that from non-senescent cells (**Fig 2A**), supporting the existence of an exosome/EV SASP (eSASP) in addition to the sSASP, as described [26]. Most changes in the fibroblast sSASP, independent of inducer, exhibited increased secretion by senescent cells, with only 1%–6% of proteins secreted at lower levels. In contrast, one half to two thirds of all significant protein changes in exosomes/EVs from senescent fibroblasts declined relative to quiescent cells (**Fig 2A**).

We also measured the secretion of known SASP factors (**S1D Fig**) in the fibroblast sSASP and eSASP, as well as the renal epithelial cell sSASP. These factors included CXCLs, high mobility group box 1 protein (HMGB1), insulin-like growth factor binding proteins (IGFBPs), matrix metalloproteinases (MMPs), laminin subunit beta-1 (LAMB1), and tissue inhibitors of

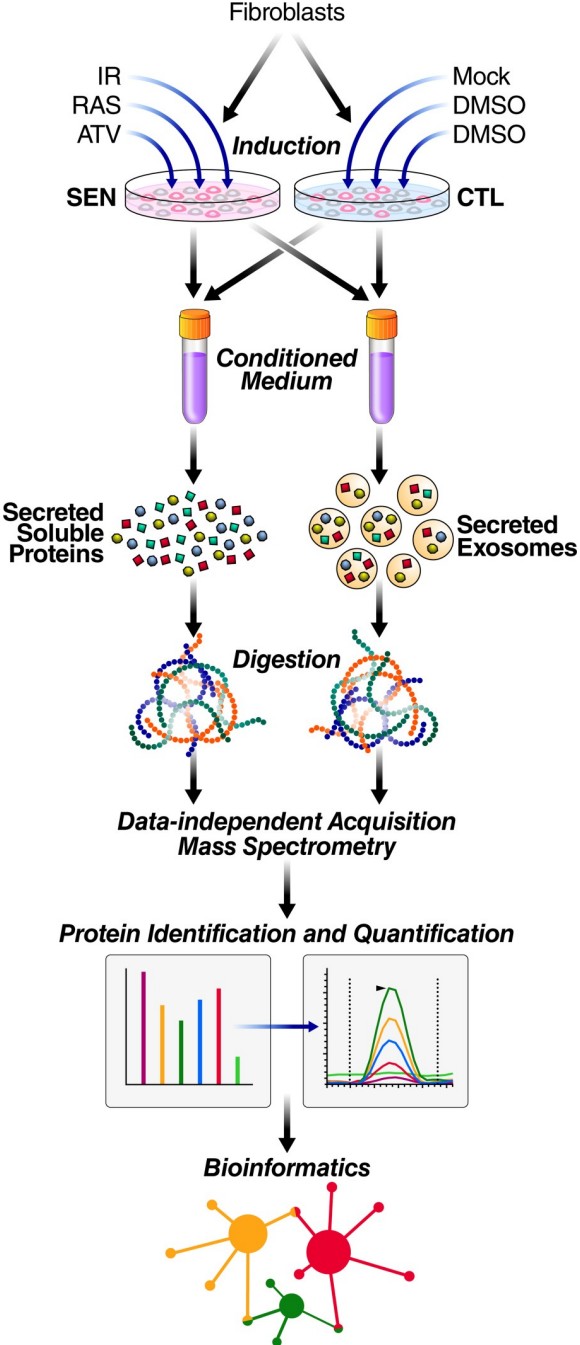

**Fig 1. Proteomic workflow for isolation and analysis of secreted proteins and exosomes/EVs.** Senescence was induced in cultured primary human lung fibroblasts by either IR, RAS, or ATV. Quiescent control cells were either mock irradiated or vehicle treated. Soluble proteins and exosomes/EVs were then isolated from conditioned media. Samples were digested and subjected to mass spectrometric analysis (DIA), followed by protein identification and quantification using Spectronaut Pulsar [32] and by bioinformatic, pathway, and network analyses in R and Cytoscape [33,34]. ATV, atazanavir treatment; CTL, Control; DIA, data-independent acquisition; EV, extracellular vesicle; IR, X-irradiation; RAS, inducible RAS overexpression; SEN, senescent.

metallopeptidase (TIMPs). In fibroblasts, nearly all previously identified SASP factors were elevated, regardless of the senescence inducer. However, while expression of p16$^{INK4a}$, IL-6, and SA-β-Gal were also elevated in renal epithelial cells (**S1A, S1B and S1C Fig**), several SASP

## A  Fibroblast Secretome/SASP

| | Soluble SASP | | | Exosomes | | |
| --- | --- | --- | --- | --- | --- | --- |
| | All | | Changed | All | | Changed |
| **IR** Genotoxic | 1505 Proteins Identified | | **548↑ 37↓** | 1502 Proteins Identified | | **180↑ 320↓** |
| **RAS** Oncogenic | 1693 Proteins Identified | | **704↑ 10↓** | 354 Proteins Identified | | **21↑ 18↓** |
| **ATV** Treatment | 441 Proteins Identified | | **332↑ 8↓** | ↑ Significantly Increased (q < 0.05, 1.5-fold change)  ↓ Significantly Decreased (q < 0.05, 1.5-fold change) | | |

## B  Pathways Increased in the Fibroblast "Core" sSASP

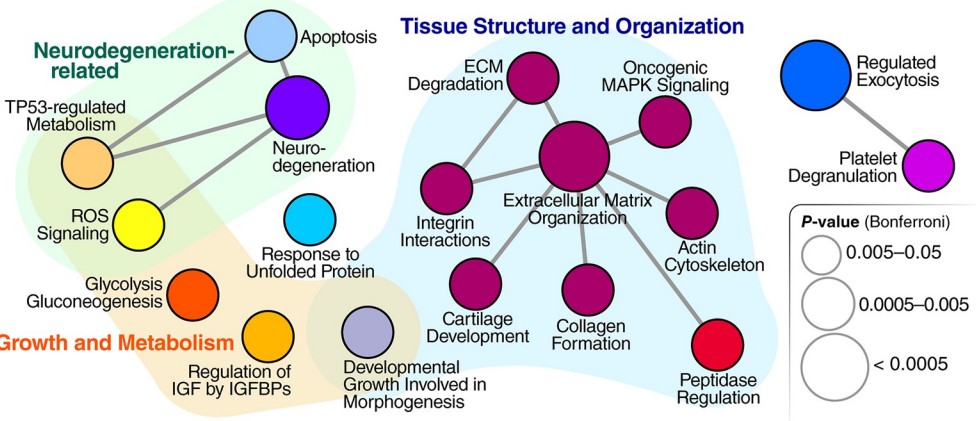

## C  Fibroblast sSASP (Increased)

## D  Fibroblast sSASP (Increased)

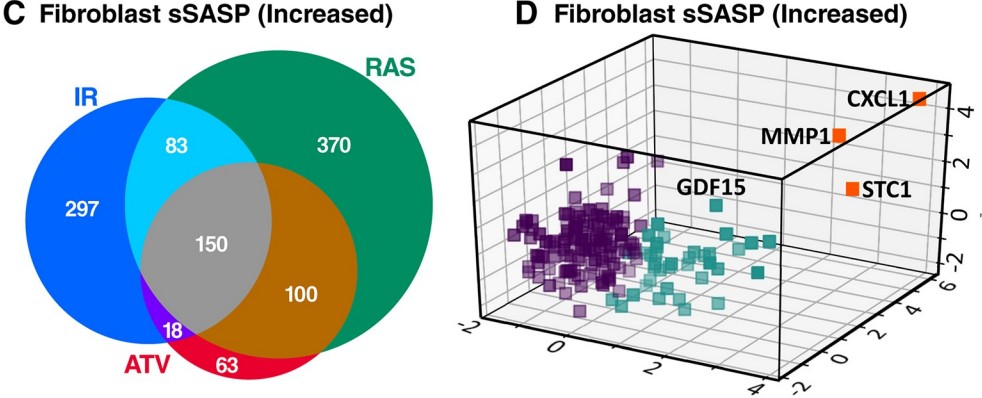

**Fig 2. Core sSASP proteins, networks and pathways.** (A) Summary of proteins with significantly altered (q-value <0.05 and >1.5-fold change) secretion by senescent compared with quiescent cells following genotoxic, oncogenic, or ATV treatment stress in senescent human lung fibroblasts. (B) ClueGO [33] pathway enrichment and network analyses of overlapping sSASPs resulting from each senescence inducer. Pathways of the same color have ≥50% similarity. Connecting lines represent Kappa connectivity scores >40%. (C) Venn diagram of proteins showing significantly increased secretion in senescent versus non-senescent fibroblasts following induction of senescence by IR, RAS, or ATV. (D) Unsupervised K-means clustering of proteins significantly increased in the sSASPs of all inducers based on the magnitude of the protein changes (log2-fold change) in senescent versus control groups and partitioned into three clusters. ATV, atazanavir treatment; CXCL1, chemokine C-X-C motif ligand 1; ECM, extracellular matrix; GDF15, growth/differentiation factor 15; IGF, insulin-like growth factor; IGFBP, IGF binding protein; IR, X-irradiation; MMP1, matrix metalloproteinase-1; RAS, RAS oncogene overexpression; ROS, reactive oxygen species; sSASP, soluble senescence-associated secretory phenotype; STC1, stanniocalcin 1; TP53, tumor protein p53.

proteins identified in fibroblasts were either decreased or unchanged, except for IGFBPs 2/3 and CXCL8. This finding suggests that fibroblast SASP markers do not necessarily pertain to other cell types. Similarly, within exosomes/EVs secreted by senescent fibroblasts, several previously identified key SASP factors were either absent, unchanged, or decreased, including

IGFBPs 2/3/5 and LAMB1, and none were consistently elevated in response to more than one inducer (S1D Fig).

### Senescence-inducing stimuli drive largely distinct secretory phenotypes

To determine how different senescence-inducing stimuli affect the SASPs, we compared the sSASP from human primary fibroblasts induced to senesce by IR, RAS, and ATV. Strikingly, the sSASP was largely distinct among inducers, with an overlap of 150 proteins among 1,091 total increased proteins and no overlap among decreased proteins (S2 Table). Thus, most sSASP protein components and corresponding changes were highly heterogeneous and not shared among inducers (Fig 2C).

To determine whether there are core pathways associated with the sSASPs, we performed pathway and network analyses on overlapping proteins in the sSASPs of each inducer (Fig 2B). The largest pathway associated with all inducers related to tissue and cell structure, including extracellular matrix organization, actin cytoskeleton, integrin interactions, and peptidase regulation.

To distill the overlapping "core" sSASP proteins into primary components, we performed an unsupervised machine learning analysis (Fig 2D). K-means clustering analysis uncovered three primary clusters among core sSASP components. Strikingly, one cluster, consisting of just three proteins—chemokine C-X-C motif ligand 1 (CXCL1), MMP1, and stanniocalcin 1 (STC1)—were highly represented in the sSASPs of all inducers, suggesting these proteins might serve as surrogate markers of the sSASP. Of note, STC1, among the top sSASP proteins, is a previously unidentified SASP factor and a secreted hormone with many disease associations [35–40]. Our analyses also validate MMP1 and CXCL1 as SASP markers.

We also generated proteomic sSASP signatures that were exclusive to two of the senescence inducers, IR and RAS (Fig 3A and 3B), to explore whether we could predict the originating inducer of senescence in published data. Due to the relative scarcity of published proteomic analyses of the sSASP, we compared our secretome analysis to published transcriptomic data. We prepared unique proteomic signatures of IR and RAS from our own data by filtering for secreted proteins that were exclusively increased in the secretomes of either IR- or RAS-induced senescent fibroblasts (S7 Table), and compared with published transcriptomic signatures of senescent cells for both stimuli [24,41–43]. The meta-transcriptomic profiles, as described by Hernandez-Segura and colleagues [24], are based on whole transcriptome (at least three replicates each) profiles of several fibroblast strains: MRC-5 and HFF datasets were used for analysis of IR-induced senescence and IMR-90 were used for RAS-induced senescence. We combined these analyses with a more recent RNA-sequencing analysis of both IR-induced senescence in WI-38 and IMR-90 human fibroblasts and RAS-induced senescence in WI-38 fibroblasts [44] to generate lists of genes that are exclusively expressed in senescent fibroblasts following IR or RAS (S7 Table). The combined transcriptome profiles contained 33 gene expression changes exclusive to IR-induced senescence and 1,749 gene expression changes exclusive to RAS-induced senescence. Likely due to the small overlap in IR-specific genes across transcriptomic studies, there was no overlap between IR-specific sSASP factors identified in our study and IR-specific transcriptome changes in senescent cells. However, of the 1,749 exclusively RAS-specific transcriptomic signature genes, we identified 26 proteins were exclusively secreted by RAS-induced senescent fibroblasts in this study (Fig 3C and S7 Table), so we focused on generating a RAS-specific signature. The RAS signature included five proteins—CXCL5, MMP9, MMP3, Cystatin-S (CST4), and C-C motif chemokine 3 (CCL3)—that were both robustly increased in both the secretomes (log2-fold change between 2.8 and 6.9) and transcriptomes (log2-fold change between 4.9 and 9.1) (Fig 3D).

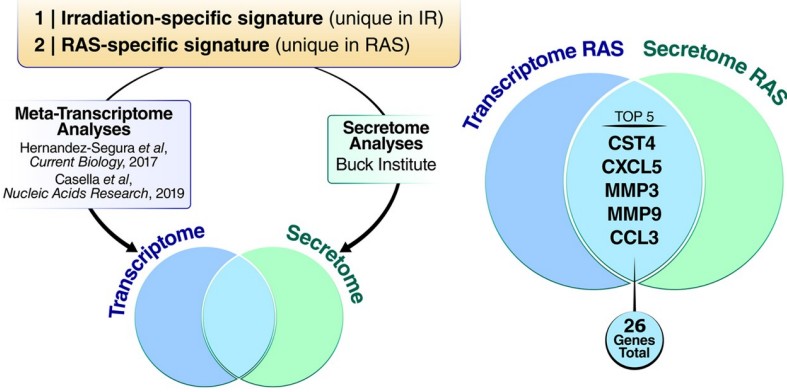

**A** Top 10 Secreted Proteins
−Exclusive to IR Secretomes−

| Genes | Log2(SEN/CTL) |
| --- | --- |
| SRP72 | 12.07 |
| FBL | 10.40 |
| CLASP1 | 10.36 |
| KPNA1 | 9.93 |
| TRAPPC6B | 9.78 |
| CAMK2B | 8.98 |
| ESYT2 | 8.76 |
| TUFM | 8.18 |
| RSF1 | 8.11 |
| CLIPX | 7.90 |

**B** Top 10 Secreted Proteins
−Exclusive to RAS Secretomes−

| Genes | Log2(SEN/CTL) |
| --- | --- |
| MAP2 | 7.33 |
| CXCL5 | 6.91 |
| SEMA7A | 6.75 |
| MMP9 | 6.57 |
| GERPL1 | 6.33 |
| HADHA | 6.03 |
| CYFIP2 | 5.63 |
| PVR | 5.50 |
| S100A6 | 5.22 |
| RPS18 | 5.16 |

**C** Inducer-specific RNA/Protein Signatures

1 | Irradiation-specific signature (unique in IR)
2 | RAS-specific signature (unique in RAS)

Meta-Transcriptome Analyses
Hernandez-Segura et al, Current Biology, 2017
Casella et al, Nucleic Acids Research, 2019

Secretome Analyses
Buck Institute

Transcriptome    Secretome

Transcriptome RAS    Secretome RAS

TOP 5
CST4
CXCL5
MMP3
MMP9
CCL3

26 Genes Total

**D** Top RAS-specific RNA/protein Signatures

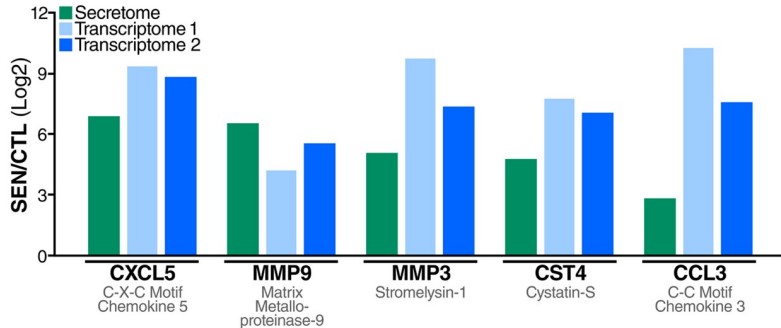

■ Secretome
■ Transcriptome 1
■ Transcriptome 2

SEN/CTL (Log2)

| CXCL5 | MMP9 | MMP3 | CST4 | CCL3 |
| --- | --- | --- | --- | --- |
| C-X-C Motif Chemokine 5 | Matrix Metallo-proteinase-9 | Stromelysin-1 | Cystatin-S | C-C Motif Chemokine 3 |

**Fig 3. Inducer-specific RNA and protein signatures of the sSASP.** Tables of top 10 inducer-specific sSASP proteins in (A) IR-induced senescence and (B) RAS-induced senescence. (B) Workflow for generating inducer-specific RNA and protein signatures of senescent cells. Transcriptome analysis of IR- and RAS-induced senescent fibroblasts were obtained from published studies [24,44] and combined. Transcriptome data were filtered for changes that were inducer specific (genes changing exclusively in one inducer but not the other) and were consistent in both studies. Inducer-specific transcriptomes were then compared with inducer-specific secretome changes in the sSASP (from the current study) to produce a combined inducer-specific RNA and protein signature. (D) Log2-fold changes of the top five RAS-specific genes in the sSASP secretome and in two published transcriptome datasets [24,44]. CTL, quiescent control; IR, X-irradiation; RAS, inducible RAS overexpression; SEN, senescent; sSASP, soluble senescence-associated secretory phenotype.

## sSASP is largely distinct in composition and regulation in fibroblasts and epithelial cells

We compared the secretomes of radiation-induced senescent lung fibroblasts and similarly treated senescent renal epithelial cells to determine the cell-type specificity of the sSASP. For renal epithelial cells, the sSASP comprised a mixture of proteins with significantly lower or

higher relative secretion (60% increased, 40% decreased), whereas 94% of protein changes in fibroblasts increased in secretion. The magnitude of the fold changes in the sSASP were significantly higher in fibroblasts than in renal epithelial cells, regardless of inducer (S4 Fig, $p < 0.0001$). For example, 531 of significant protein changes in the fibroblast sSASP were >2-fold, compared to 138 in the renal epithelial cell sSASP. However, for renal epithelial cells, an additional 212 proteins showed significant changes between 1.5- and 2-fold increase or decrease.

The sSASP of irradiated fibroblasts and epithelial cells were largely distinct (Fig 4A, 4B and 4C). Among the proteins increased in the sSASP of each cell type, 9%–23% overlapped, and the magnitude of the changes by renal epithelial cells were, in most cases, lower than in fibroblasts regardless of the senescence inducer, although it is possible that senescent fibroblasts secrete more protein overall than epithelial cells in response to stress. Interestingly, 20%–30% of proteins significantly decreased in the sSASP of renal epithelial cells overlapped with proteins significantly increased in the fibroblast sSASP (Fig 4B). Among the epithelial factors that changed oppositely to the fibroblast factors were IGFBPs 4/7, TIMPs 1 and 2, CXCL1, and most serine protease inhibitors (SERPINs). In all, 17 sSASP factors were shared between all senescence inducers and cell types we examined (S3 Table).

Pathway and network analysis of proteins increased in the sSASPs of epithelial cells (Fig 4C) showed that most pathways belonged to one of three general categories: protein turnover and secretion, primary metabolism, and cellular detoxification. While not as apparent on a molecule-by-molecule basis, many pathways were commonly enriched in both the epithelial and fibroblast sSASPs (Figs 4D and 2B), including vesicle-mediated transport and exosomes, glycolytic metabolism, and cellular detoxification. Of notable exceptions, pathways enriched uniquely by epithelial cells included protein translation and degradation (lysosome and phagosome).

Surprisingly, most renal epithelial sSASP proteins with significantly lower secretion by senescent cells were enriched in pathways related to tissue and cell structure, adhesion, and motility (Fig 4E). This finding contrasts with previous reports and our own analyses of fibroblasts (Fig 2B), in which these pathways were increased regardless of inducer. The irradiated epithelial sSASP also had significantly lower levels of proteins involved in RNA processing, in contrast to increased RNA metabolism in the irradiated fibroblast sSASP. Additionally, the epithelial sSASP was significantly depleted in proteins related to proteasome degradation, antigen processing, and the complement system.

Damage-associated molecular patterns (DAMPs; also known as alarmins or danger signals) are released from cells in response to internal and external stress, and are components of the sSASP [45]. HMGB1 is a founding member of the DAMPs, a prominent sSASP marker, and, along with calreticulin (CALR), an important driver of inflammation [45]. Our analysis identified increased secretion of multiple DAMPs, including HMGB1 and CALR, by senescent fibroblasts under all senescence inducers (Table 1). However, with some exceptions, the secretion of central DAMPs was unchanged or significantly reduced by senescent renal epithelial cells, demonstrating that some defining sSASP components vary depending on cell type.

## Exosome/EV proteomic signatures are altered by cellular senescence

Because proteins are also secreted as EV cargo, we hypothesized that senescent cells would show significant changes in this fraction, which we term the exosome/EV SASP (eSASP). We used ultracentrifugation to enrich conditioned media for exosomes and small EVs released by quiescent and senescent fibroblasts induced by IR and RAS (Fig 1). We confirmed the quality of exosome/EV–purified fractions by measuring the presence of multiple EV-specific markers,

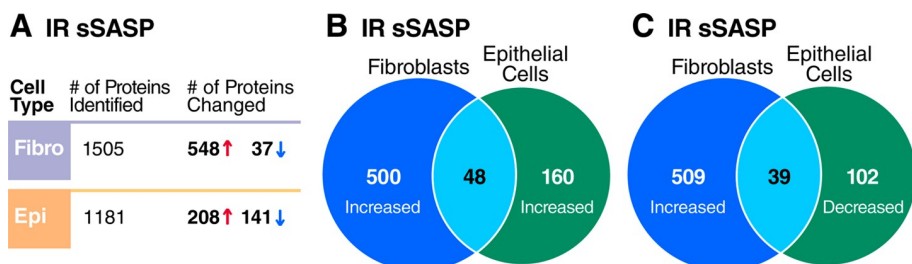

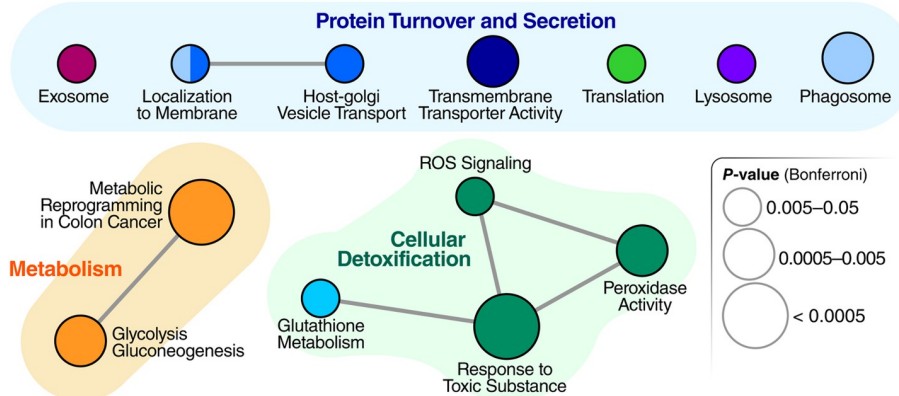

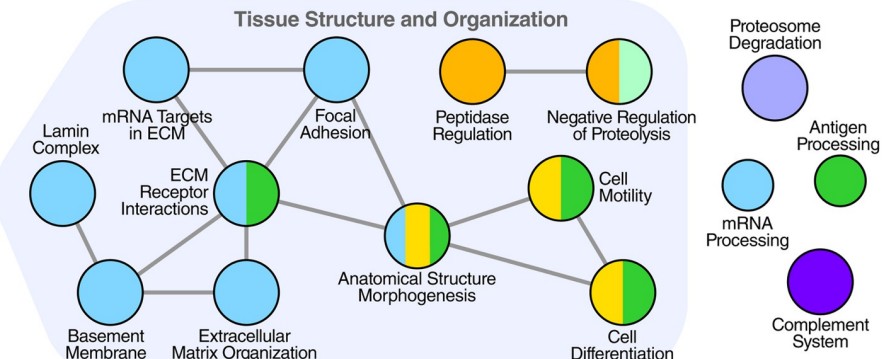

**Fig 4. Epithelial cells and fibroblasts exhibit distinct sSASPs.** (A) Number of proteins identified and significantly altered in the sSASP of irradiated fibroblasts and epithelial cells. (B) Venn diagram comparing proteins significantly increased in the sSASPs of senescent fibroblasts and epithelial cells, both induced by IR (q < 0.05). (C) Venn diagram comparing protein increases in the fibroblast sSASP versus decreases in the epithelial sSASP. (D) Pathway and network analysis of secreted proteins significantly increased in epithelial cell sSASP. (E) Pathway and network analysis of proteins significantly decreased in the epithelial cell sSASP. ECM, extracellular matrix; IR, X-irradiation; ROS, reactive oxygen species; sSASP, soluble senescence-associated secretory phenotype.

including CD63, CD9, CD81, and CDC42 [46], in our proteomic data (**S5A Fig**), by independent antibody-based detection of 37 exosomal surface isotopes (**S5D and S5E Fig**) and by particle counting and size distribution analysis (**S5B and S5C Fig**).

To determine whether the characteristics of exosomes/EVs from senescent cells are altered, we analyzed particle number and size distribution of exosomes/EVs secreted into the culture medium of senescent and non-senescent cells over a 24-hour period. On average, senescent cells released a greater number of vesicles—about 68 per cell compared with 50 per control cell (**S5B Fig**). The mean diameter of senescent exosomes/EVs (147 nm) was 2% lower ($p < 0.001$)

**Table 1. DAMPs are a core component of the fibroblast sSASP.**

| | Log2(SEN/CTL) | | | |
|---|---|---|---|---|
| | **IR (Fibroblasts)** | **RAS** | **ATV** | **IR (Epithelial)** |
| **HMGB1** | 2.47 | 0.59 | 2.46 | NS |
| **CALR** | 1.22 | 0.51 | 1.32 | −0.75 |
| **CD44** | 2.25 | 1.20 | 1.92 | −0.51 |
| **S100A11** | 0.56 | 1.35 | 1.88 | 0.64 |
| **LGALS3BP** | 1.46 | 1.76 | 1.79 | −1.14 |
| **VCAN** | 1.80 | 1.32 | 0.98 | −1.39 |
| **TNC** | 1.64 | 1.40 | 2.46 | 0.39 |
| **HSPA5** | 2.03 | 3.93 | 1.78 | −0.24 |
| **HSP90AB1** | 5.01 | 2.69 | 1.65 | 0.26 |
| **HSPA8** | 2.49 | 2.98 | 1.46 | 0.49 |
| **HSPA1A** | 2.96 | 2.40 | 1.45 | 0.67 |
| **HSP90AA1** | 4.94 | 3.42 | 1.34 | NS |
| **HSP90B1** | 2.67 | 1.61 | 0.66 | NS |

All changes are significant (q < 0.05) unless denoted NS.

Abbreviations: ATV, atazanavir treatment; CALR, calreticulin; CTL, quiescent control; DAMP, damage-associated molecular pattern; HMGB1, high mobility group box 1 protein; IR, X-irradiation; NS, not significant; RAS, oncogenic RAS overexpression; SEN, senescent; sSASP, soluble senescence-associated secretory phenotype.

than quiescent control exosomes/EVs (150 nm) (**S5B and S5C Fig**). Further work using senolytics may validate whether the number, size, and other characteristics of secreted exosome/EVs are indicators of senescent cell burdens in humans.

The protein content of exosomes/EVs released by IR- versus RAS-induced senescent fibroblasts was largely distinct, sharing only 9 significantly altered proteins (**Fig 5A**). Exosomes/EVs were reported to contain protein signatures of their originating cells [28,47], offering a unique opportunity to identify senescence biomarkers with a degree of cell type specificity. Thus, exosome/EV proteins might distinguish senescent cells of different origins or resulting from different stressors. The membranes of exosomes are also representative of the originating cells [28,47]. Indeed, about 30% of all the exosome/EV proteins that increased upon senescence are plasma membrane proteins (**Fig 5B**), suggesting that exosomes/EVs might also identify cell type origins through their cell-surface proteins, although this will need to be experimentally confirmed.

Protein changes in exosomes released by senescent cells were also largely distinct from changes in soluble protein secretion obtained from matching cells. Of all 548 soluble proteins secreted at significantly higher levels in senescent fibroblasts following IR, only 51 (9.3%) significantly increased in the eSASP of senescent fibroblasts following IR, and 70 proteins (12.8%) significantly decreased in the eSASP. Accordingly, the pathways enriched within eSASP were largely distinct from the sSASP of irradiated fibroblasts. In addition to enrichment of proteins involved in membrane organization, such as cell adhesion and cell junction assembly proteins, the eSASP is uniquely enriched with signaling pathways such as RAS signaling, G-protein signaling, and prostaglandin synthesis and regulation (**Fig 5C**), as opposed to the metabolic, growth, and extracellular matrix (ECM)-remodeling pathways found in the cell and inducer-matched sSASP. Full lists of proteins secreted by senescent exosomes are in **S1 Table**.

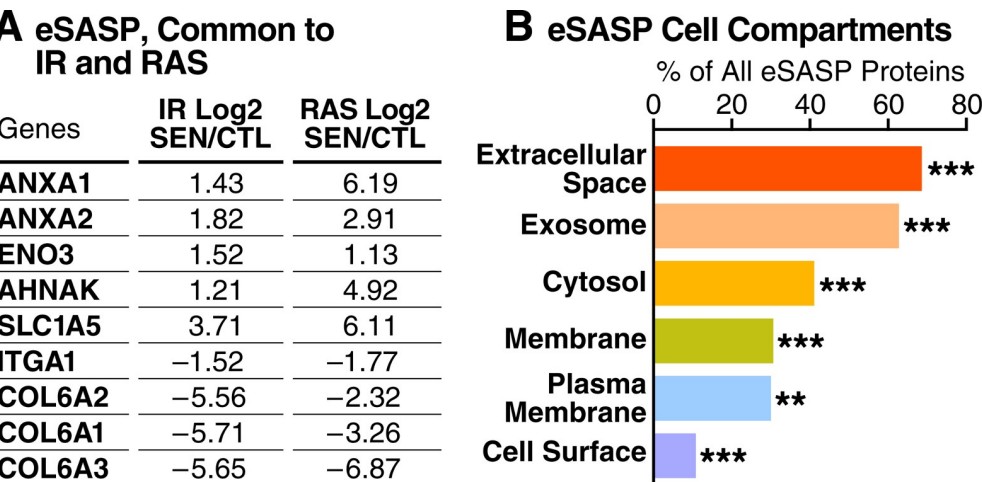

## A eSASP, Common to IR and RAS

| Genes | IR Log2 SEN/CTL | RAS Log2 SEN/CTL |
|---|---|---|
| ANXA1 | 1.43 | 6.19 |
| ANXA2 | 1.82 | 2.91 |
| ENO3 | 1.52 | 1.13 |
| AHNAK | 1.21 | 4.92 |
| SLC1A5 | 3.71 | 6.11 |
| ITGA1 | −1.52 | −1.77 |
| COL6A2 | −5.56 | −2.32 |
| COL6A1 | −5.71 | −3.26 |
| COL6A3 | −5.65 | −6.87 |

## B eSASP Cell Compartments

% of All eSASP Proteins

## C Pathways Increased in Irradiated Fibroblast eSASP

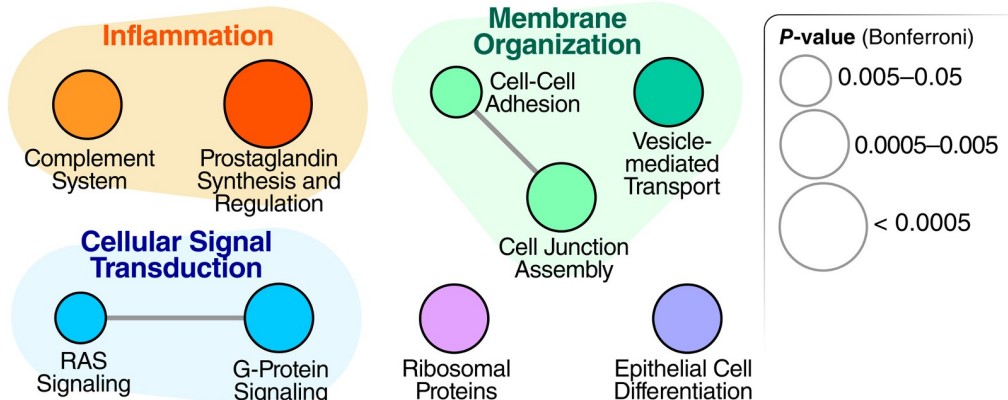

**Fig 5. Cellular senescence alters exosome/EV features and composition.** (A) Table showing overlapping significant protein changes in exosomes/EVs secreted by senescent cells induced by IR versus RAS (q < 0.05). (B) Enrichment analysis of gene-ontology/cellular compartments overrepresented among protein contents of exosomes/EVs released by senescent cells. (C) Network analysis of pathways and functions unique to the eSASP. CTL, quiescent control; eSASP, extracellular vesicle senescence-associated secretory phenotype; EV, extracellular vesicle; IR, X-irradiation; RAS, inducible RAS overexpression; SEN, senescent.

## The sSASP contains potential aging and disease biomarkers

As a driver of many aging and disease phenotypes, the sSASP could include known biomarkers of aging and age-related diseases. A recent biomarker study identified 217 proteins that are significantly associated with age in human plasma (adjusted $p < 0.00005$) [48]. Of these, 20 proteins (9.2%) were present in the originally defined SASP [2]. Strikingly, multiple newly identified SASP factors from our present study were also identified in the study of human plasma [48] (**Fig 6**). Of all the originally defined sSASP factors and unique sSASP proteins that we identify here, 92 proteins were also identified as markers of aging in human plasma (42.4% of all plasma aging markers) (**Fig 6A, 6C and 6D** and **S4 Table**). Considering the originally defined SASP in addition to our newly identified "core SASP" (sSASP components common to all senescence inducers), the number of age-associated plasma proteins that are also sSASP proteins is 39, or 18.0% of plasma aging markers (**Fig 6B, 6C and 6D** and **S4 Table**). Thus, plasma biomarkers of aging are highly enriched with sSASP factors.

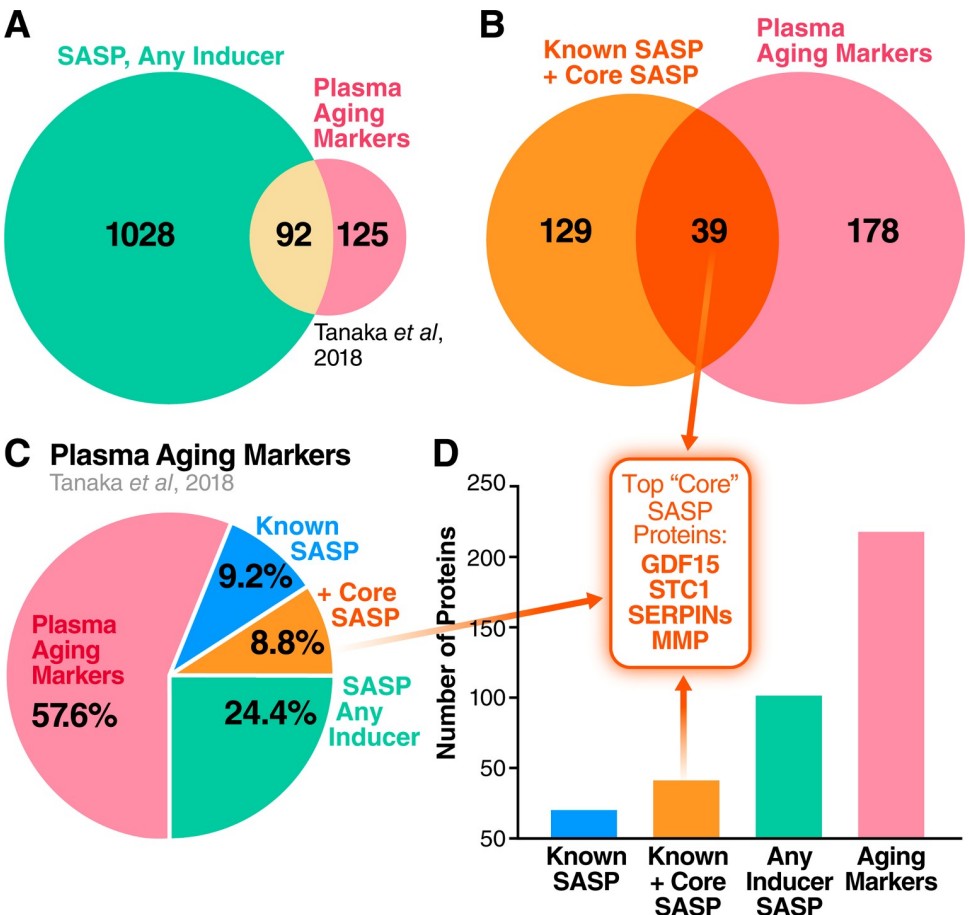

**Fig 6. Human plasma aging markers are enriched for sSASP proteins.** (A) Venn diagram comparing sSASP factors secreted by at least one of IR-, RAS-, or ATV- induced senescent cells with markers of aging identified in human plasma [48]. (B) Overlap between the core sSASP (proteins secreted following all senescence-inducing stimuli) and plasma aging markers. (C) Pie chart showing the proportion of known sSASP factors, newly identified core sSASP factors, and sSASP factors found among plasma markers of aging in humans. (D) Number of proteins contained in the originally identified sSASP, core sSASP, noncore sSASP, and markers of aging in human plasma [48] ($p < 0.00005$). Top core sSASP factors GDF15, STC1, SERPINs, and MMP1 are among the plasma aging markers. ATV, atazanavir treatment; GDF15, growth/differentiation factor 15; IR, X-irradiation; MMP, matrix metalloproteinase; RAS, inducible RAS overexpression; SASP, senescence-associated secretory phenotype; SERPIN, serine protease inhibitors; sSASP, soluble senescence-associated secretory phenotype; STC1, stanniocalcin 1.

Complement and coagulation cascade proteins [18], particularly protease inhibitors such as SERPINs, were also noted as prominent plasma biomarkers of aging [48]. These proteins and their pathway networks were robustly altered in the sSASPs of cells induced to senesce by all the tested stressors (**Figs 2B and 7A**), in addition to other top biomarker candidates: MMP1, STC1, and GDF15 (**Fig 7B, 7C and 7D**). The protein having the strongest association with aging [48], GDF15 (r = 0.82), was among the most highly secreted proteins in the sSASP induced by IR, RAS and ATV in fibroblasts, and in epithelial cells induced by IR (**Fig 7D**). Increased secretion of top core sSASP biomarkers plasminogen activator inhibitor 1 (SER-PINE1), MMP1, STC1, and GDF15 was confirmed by western blotting in RAS-induced senescent cells compared to controls (**S3 Fig**). The enrichment of aging and disease biomarkers in the secretomes of senescent cells supports their link to a wide spectrum of age-related diseases.

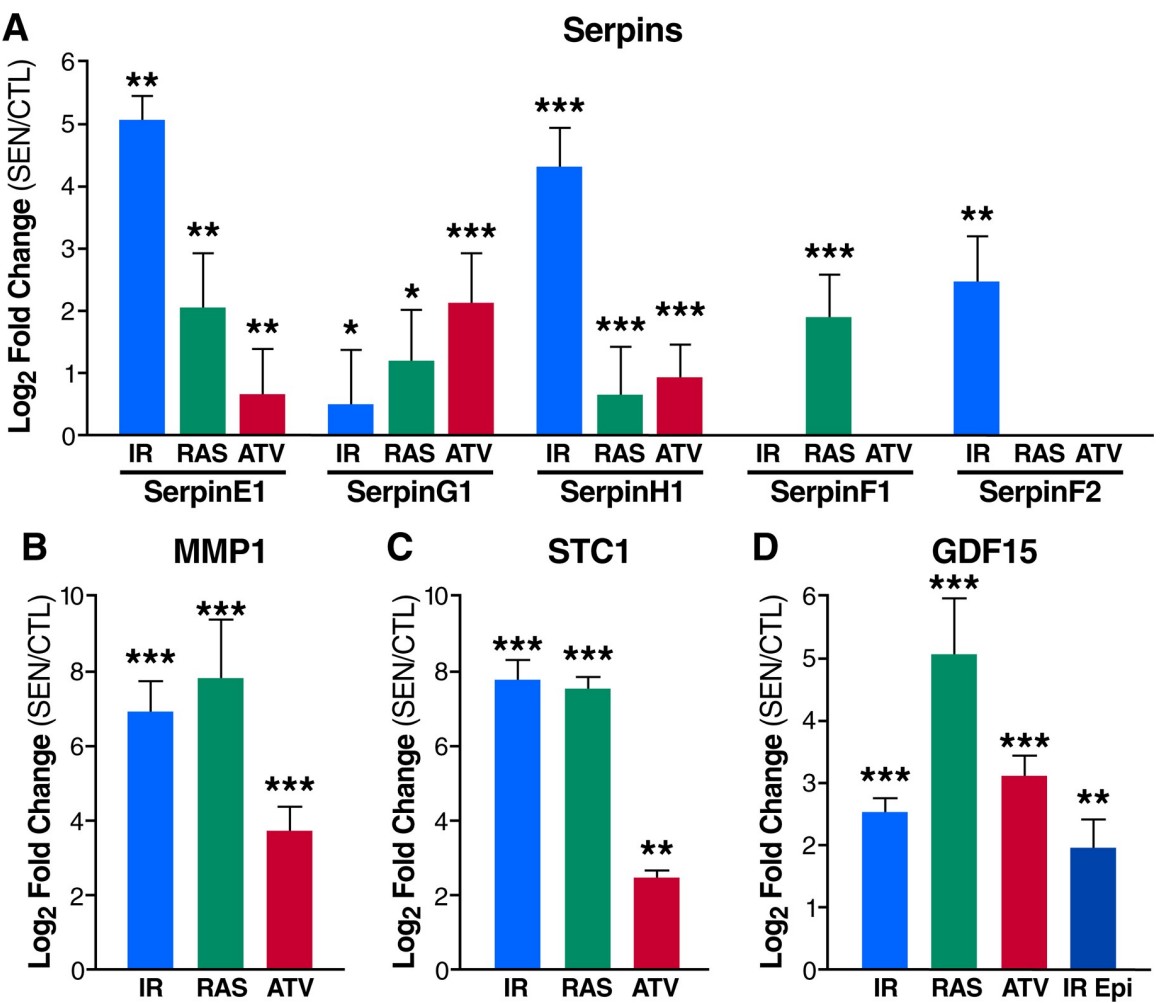

**Fig 7. The sSASP contains aging and disease biomarkers.** (A) Serpins are secreted at high levels by senescent fibroblasts induced by IR, RAS, or ATV. (B) MMP1 and (C) STC1 are among the most highly secreted proteins by senescent fibroblasts. (D) The plasma aging biomarker GDF15 is increased in the sSASPs of fibroblasts induced to senesce by IR, RAS, and ATV and epithelial cells induced by IR. $^*q < 0.05$, $^{**}q < 0.01$, $^{***}q < 0.001$. ATV, atazanavir treatment; CTL, quiescent control; Epi, renal epithelial cell; GDF15, growth/differentiation factor 15; IR, X-irradiation; MMP1, matrix metalloproteinase-1; RAS, inducible RAS overexpression; SEN, senescent; sSASP, soluble senescence-associated secretory phenotype; STC1, stanniocalcin 1.

## Discussion

Here, we present the SASP Atlas (www.SASPAtlas.com), the first proteome-based database of SASPs. This database contains the contents of exosome/EV and soluble secretomes, in addition to SASPs originating from multiple senescence-inducing stresses and two distinct cell types. The SASP Atlas will be continuously updated with SASP profiles from new cell types and senescence, including paracrine (or bystander) senescence [49,50] as well as temporal dynamics of the SASP—all generated by our laboratories.

Our proteomic analysis leverages a modern DIA mass spectrometry workflow, which comprehensively acquires label-free, quantitative peptide (MS1), and fragment-level (MS2) data for all peptides in each sample [30–32,51,52]. DIA workflows are not limited by the stochastic peptide MS2 sampling biases characteristic of traditional data-dependent acquisition (DDA) mass spectrometry. In addition to the SASP Atlas database, we provide panels of SASP factors on Panorama Web, a freely available web repository for targeted mass spectrometry assays

[53,54]. These resources can be used as a reference and guide to identify and quantify SASP factors that may be associated with specific diseases, and to develop aging and disease-related biomarkers (**Fig 8**).

SASP profiles are needed to develop senescence biomarkers in human plasma or other biofluids, and for identifying individuals to treat with, and measuring the efficacy of, senescence-targeted therapies such as senolytics. Translating senescence- and SASP-targeted interventions to humans will require a comprehensive profile of SASPs, both to identify their deleterious components and to develop human biomarkers to assess senescent cell burden. The SASP, as originally identified, comprised approximately 50 cytokines, CXCLs, growth factors, and

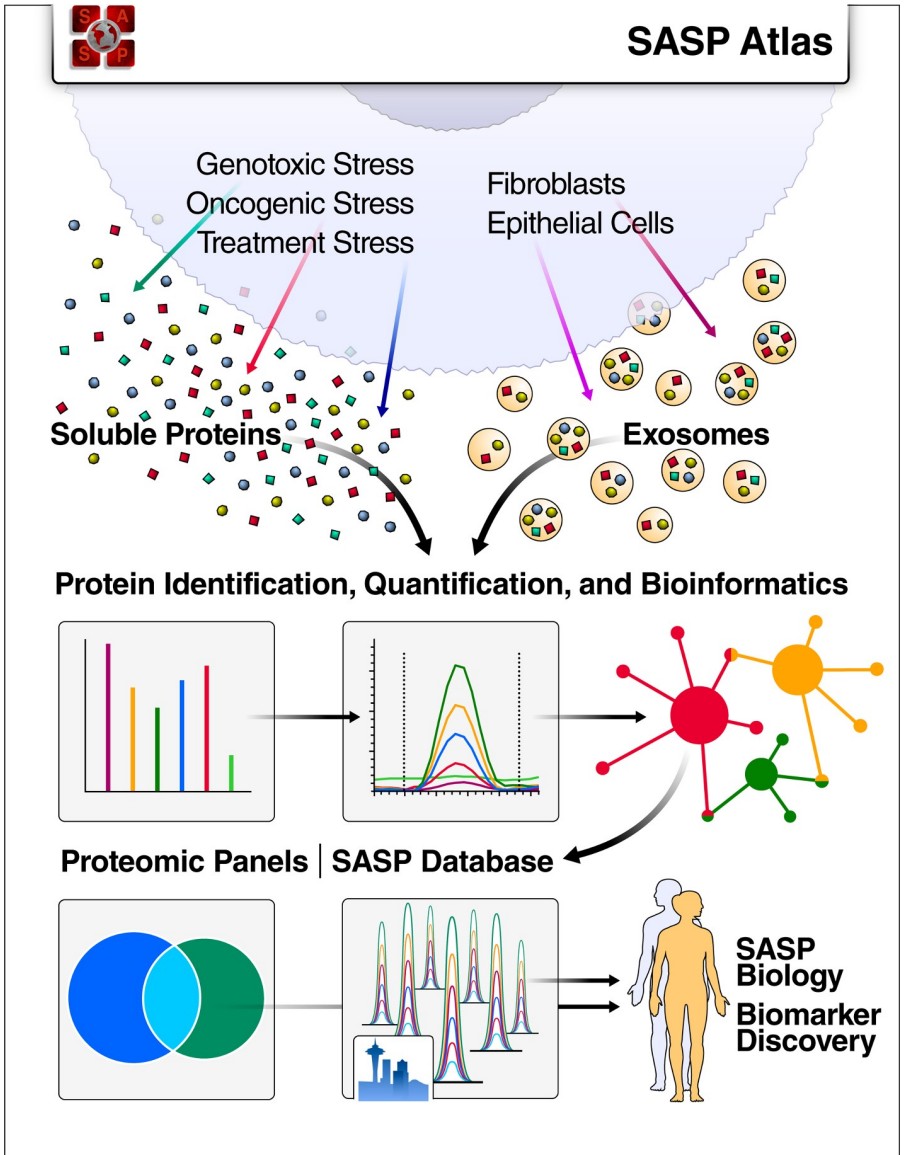

**Fig 8. SASP Atlas: A comprehensive resource for SASPs.** SASP Atlas ([www.SASPAtlas.com](www.SASPAtlas.com)) is a curated and freely available database of the secretomes of senescent cells, including both the soluble and exosome SASP, that can be used to identify SASP components or biomarker candidates for senescence burden, aging, and related diseases. SASP, senescence-associated secretory phenotype.

proteases that were detected by biased methods (e.g., antibody arrays) and/or transcriptional analyses [1–4,24]. While these comprehensive analyses are valuable in describing the overall phenotype of senescent cells, proteomic analyses are complementary in both confirming transcriptional changes and identifying and quantifying novel SASP factors that are not apparent at the mRNA level. For example, a recent meta-analysis of senescent cell transcriptomes [24] identified >1,000 genes with increased expression specifically in senescent cells induced by IR or oncogenic RAS, and >700 "core" senescence genes (increased expression following all senescence inducers tested). Our analysis identified 548, 644, and 143 proteins in the IR, RAS, and core sSASP, respectively, that were previously unreported at the RNA level (**S6 Fig**). We expect that the number and nature of these sSASP core proteins will change as we and others interrogate additional cell types and senescence inducers, and we will continue to curate the interactive SASP Atlas. Additionally, the secretion of sSASP factors, such as HMGB1 and other DAMPs, is not generally transcriptionally driven.

DAMP receptor–bearing cells, including cells of the innate immune system, recognize extracellular DAMPs as signals to promote inflammatory and fibrotic responses. Increased circulating DAMPs are hypothesized to play a role in aging [55,56], particularly the age-related inflammation termed "inflammaging" [57]. DAMPs can also serve as biomarkers of a number of diseases, including trauma and cardiovascular, metabolic, neurodegenerative, malignant, and infectious diseases [55,58,59]. In addition, our top "core sSASP" biomarker candidates have been identified as disease biomarkers in human studies. For example, human cohort studies have reported GDF15 as a biomarker of cardiovascular disease, cardiovascular and cancer mortality and morbidity, renal disease, and all-cause mortality independent of cardiovascular mortality [60–66], as well as a driver of senescence-associated colon cancer metastasis [67]. Additionally, two of the top core sSASP proteins identified by an unbiased k-means clustering algorithm—STC1 and MMP1 (**Fig 7B and 7C**)—were reported as significant aging biomarkers [48]. In addition to aging, MMP1 has been identified as a biomarker for several cancers, pulmonary fibrosis, and potentially Alzheimer's disease [68–71], whereas STC1 has been identified as a diagnostic and prognostic biomarker for cancers, pulmonary fibrosis, renal ischemia/reperfusion injury, and Alzheimer's disease [35–40].

It is important to note that while this study examines the SASP following several senescence-inducing stimuli, the originating stresses that drive an increasing senescent cell burden in vivo during aging and age-related diseases are unknown. We therefore focused our biomarker candidates on core soluble SASP components, which are presumably contributed by senescent cells originating from multiple stimuli. However, it is likely that this list of core components will decrease as additional stimuli and cell types are added to the SASP Atlas. Therefore, as senescent cells are better characterized in vivo, it will be important to tailor these biomarker candidates to better represent the originating cell types and stresses of senescent cells in the context of aging and specific diseases.

Inducer-specific biomarkers will be critical in cases in which the originating stimuli is known, such as in treatment-induced senescence (i.e., patients taking atazanivir or chemotherapeutics). Senescent cells also have a unique combination of changes in proteasome activity, autophagic flux, and metabolic state depending on their originating stimuli, which have been proposed as inducer-specific biomarkers [72]. We also built inducer-specific senescence signatures based on our proteomic analysis and publicly available transcriptomic data (**Fig 3** and **S7 Table**), and a combined signature for RAS-induced senescence. We propose these signatures will continue to evolve as other senescence-inducing stimuli are studied, and may serve as valuable biomarkers to identify the originating inducers of senescent cells in vivo. We will expand our efforts to build unique molecular profiles to distinguish between senescent cell types, or a subset of senescent cell types, in addition to pursuing a "core" signature.

Surprisingly, many classical features associated with the sSASP, some of which were originally described in cultured fibroblasts, were not present or even changed oppositely in renal epithelial cells. Of note was an absence or reduced secretion of DAMPS, including HMGB1 and CALR, danger signals that initiate inflammatory responses and immune clearance at sites of tissue stress and damage. Epithelial cells also showed significantly reduced secretion of proteins in pathways related to tissue structure and organization that was significantly increased in fibroblasts, including ECM organization, focal adhesion, and other pathways enriched with ECM proteins, proteases, and protease inhibitors. These differences between lung fibroblasts and renal epithelial cells highlight the importance of cell type origin on the heterogeneous senescent phenotype and suggest that the SASP is individualized to cell type. Further characterization of the SASP from both fibroblasts and epithelial cells originating in other tissues will be needed to determine whether these cell-type differences in the SASP are specific to epithelial cells from the kidney or are generalized to other types of epithelial cells.

Our quantitative unbiased proteomic analysis of senescent fibroblasts and epithelial cells reveals a much larger and diverse SASP than initially reported. These SASP profiles contribute a number of new potential senescence, aging, and disease biomarkers. By virtue of this proteomic analysis of secreted proteins, SASP profiles are also ideal candidates for plasma-based biomarkers, which are enriched among secreted proteins [73]. Senescent cells were recently shown to modulate hemostasis and clotting phenotypes in plasma in vivo [18], demonstrating that senescent cells secrete bioactive factors into circulation in vivo. The use of SASP factors as biomarker candidates is further supported by our analysis, which has indicated that core SASP factors are enriched among plasma biomarkers of aging in humans. In addition to general senescence biomarkers, many proteins will likely be specific to the cell type and originating stimulus. Thus, biomarkers present in human patients in vivo will likely vary depending on the affected tissue, originating cell types, and senescence stimuli. Therefore, comprehensive quantitative profiles of the SASP under a variety of physiological conditions will provide biomarker candidates with a higher degree of selectivity to specific pathologies in humans.

## Materials and methods

### Reagents and resources

A full list of reagents and resources, including vendors and catalog numbers, is available in a Reagent and Resource Table (S5 Table). Further information and requests for resources and reagents should be directed to the Lead Contact, Birgit Schilling (bschilling@buckinstitute.org).

### Human cell culture and primary cell lines

IMR-90 primary human lung fibroblasts (ATCC, Manassas, VA; #CCL-186) were cultured in Dulbecco's Modified Eagle Medium (DMEM; Thermo Fisher Scientific, Waltham, MA; #12430–054) supplemented with penicillin and streptomycin (5,000 U/mL and 5,000 μg/mL; Thermo Fisher Scientific, Waltham, MA; #15070063) and 10% fetal bovine serum (FBS; Thermo Fisher Scientific, Waltham, MA; #2614079). Primary human renal epithelial cells (ATCC, Manassas, VA; #PCS400011) were cultured in Renal Epithelial Cell Basal Medium (Female, ATCC, Manassas, VA; #PCS-400-030). Both cell types were maintained at 37°C, 10% $CO_2$, and 3% $O_2$. Additional information about cell culture conditions, including seeding density, culture vessels, volume of medium, and final cell counts for each experiment, are available in S6 Table.

## Induction of senescence

**IR.** Senescence was induced by ionizing radiation (IR;10 Gy X-ray). Quiescent control cells were mock irradiated. Senescent cells were cultured for 10 days to allow development of the senescent phenotype, and quiescent cells were cultured in 0.2% serum for 3 days. Cells were then washed with PBS (Thermo Fisher Scientific, Waltham, MA; #10010–023) and placed in serum- and phenol red–free DMEM (Thermo Fisher Scientific, Waltham, MA; #21063–029), and conditioned media was collected after 24 hours.

**RAS overexpression.** RAS[v12] was cloned in pLVX vector (Lenti-X Tet-On; Takara Bio, Mountain View, CA; #632162) to make inducible lentiviruses, which were used to infect early passage IMR-90 cells (PD-30). Transduced cells were selected in puromycin (1 μg/mL) for 24 hours. For induction of RAS[v12], cells were treated with 1 μg/mL doxycycline in DMSO (Sigma-Aldrich, St. Louis, MO; # D9891) for 4 (early time point) or 7 days. Doxycycline was replaced after every 48 hours. Subsequently, cells were washed with PBS and placed in serum- and phenol red–free DMEM, and conditioned media was collected after 24 hours.

**Atazanivir treatment.** Cells were cultured in appropriate media containing 20 μM atazanavir, which is a clinically relevant dose, or vehicle (DMSO) for 9 (early time point) or 14 days. Subsequently, cells were washed with PBS and placed in serum- and phenol red–free DMEM, and conditioned media was collected after 24 hours.

## Isolation of secreted soluble proteins and exosomes/EVs

Proteins secreted into serum-free medium over a 24-hour period were collected. An ultracentrifugation protocol was used to separate the exosome and small EV fraction from the soluble protein fraction [74]. Briefly, conditioned medium was centrifuged at 10,000$g$ at 4°C for 30 minutes to remove debris. The supernatant was then centrifuged at 20,000$g$ at 4°C for 70 minutes to remove microvesicles, followed by ultracentrifugation at 100,000$g$ at 4°C for 70 minutes to pellet exosomes. The exosome-depleted supernatant was saved as the sSASP. The exosome pellet was then washed twice with PBS and ultracentrifuged again at 100,000$g$ at 4°C for 70 minutes before resuspending in PBS and saving as the eSASP.

## Proteomic sample preparation

**Chemicals.** Acetonitrile (#AH015) and water (#AH365) were from Burdick & Jackson (Muskegon, MI). Iodoacetamide (IAA, #I1149), dithiothreitol (DTT, #D9779), formic acid (FA, #94318-50ML-F), and triethylammonium bicarbonate buffer 1.0 M, pH 8.5 (#T7408), were from Sigma Aldrich (St. Louis, MO), urea (#29700) was from Thermo Scientific (Waltham, MA), sequencing grade trypsin (#V5113) was from Promega (San Luis Obispo, CA), and HLB Oasis SPE cartridges (#186003908) were from Waters (Milford, MA).

**Protein concentration and quantification.** Samples were concentrated using Amicon Ultra-15 Centrifugal Filter Units with a 3-kDa cutoff (MilliporeSigma, Burlington, MA; #UFC900324) as per the manufacturer instructions and transferred into 8 M urea/50 mM triethylammonium bicarbonate buffer at pH 8. Protein quantitation was performed using a BCA Protein Assay Kit (Pierce, Waltham, MA; #23225).

**Digestion.** Aliquots of each sample containing 25–100 μg protein were brought to equal volumes with 50 mM triethylammonium bicarbonate buffer at pH 8. The mixtures were reduced with 20 mM DTT (37°C for 1 hour), then alkylated with 40 mM iodoacetamide (30 minutes at RT in the dark). Samples were diluted 10-fold with 50 mM triethylammonium bicarbonate buffer at pH 8 and incubated overnight at 37°C with sequencing grade trypsin (Promega, San Luis Obispo, CA) at a 1:50 enzyme:substrate ratio (wt/wt).

**Desalting.**   Peptide supernatants were collected and desalted with Oasis HLB 30-mg Sorbent Cartridges (Waters, Milford, MA; #186003908), concentrated, and resuspended in a solution containing mass spectrometric "Hyper Reaction Monitoring" retention time peptide standards (HRM, Biognosys, Schlieren, Switzerland; #Kit-3003) and 0.2% formic acid in water.

## Mass spectrometry analysis

Samples were analyzed by reverse-phase HPLC-ESI-MS/MS using the Eksigent Ultra Plus nano-LC 2D HPLC system (Dublin, CA) combined with a cHiPLC system directly connected to an orthogonal quadrupole time-of-flight SCIEX TripleTOF 6600 or a TripleTOF 5600 mass spectrometer (SCIEX, Redwood City, CA). Typically, mass resolution in precursor scans was approximately 45,000 (TripleTOF 6600), while fragment ion resolution was approximately 15,000 in "high sensitivity" product ion scan mode. After injection, peptide mixtures were transferred onto a C18 pre-column chip (200 μm × 6 mm ChromXP C18-CL chip, 3 μm, 300 Å; SCIEX, Redwood City, CA) and washed at 2 μL/minute for 10 minutes with the loading solvent (H$_2$O/0.1% formic acid) for desalting. Peptides were transferred to the 75 μm × 15 cm ChromXP C18-CL chip, 3 μm, 300 Å (SCIEX, Redwood City, CA) and eluted at 300 nL/minute with a 3-hour gradient using aqueous and acetonitrile solvent buffers.

All samples were analyzed by DIA, specifically using variable window DIA acquisitions [75]. In these DIA acquisitions, windows of variable width (5 to 90 m/z) are passed in incremental steps over the full mass range (m/z 400–1,250). The cycle time of 3.2 seconds includes a 250-ms precursor ion scan followed by a 45-ms accumulation time for each of the 64 DIA segments. The variable windows were determined according to the complexity of the typical MS1 ion current observed within a certain m/z range using a SCIEX "variable window calculator" algorithm (more narrow windows were chosen in "busy" m/z ranges, wide windows in m/z ranges with few eluting precursor ions) [31]. DIA tandem mass spectra produce complex MS/MS spectra, which are a composite of all the analytes within each selected Q1 m/z window. All collected data were processed in Spectronaut using a panhuman library that provides quantitative DIA assays for approximately 10,000 human proteins [76].

## Cell viability assays

Cell viability was assessed with SYTOX Green Nucleic Acid Stain (Invitrogen, Carlsbad, CA; #S7020) or propidium iodide (Thermo, Waltham, MA; #P1304MP) inclusion assay. Senescent and control cells were incubated for 24 hours in serum-free medium containing SYTOX Green with continuous imaging. Cell death was quantified by counting total SYTOX Green positive nuclei (523 nm) after 24 hours. For propidium iodide inclusion assays, cells were incubated with propidium iodide for 1 hour, and cell death was quantified by counting the number of propidium iodide–positive cells (617 nm).

## SA-β-Gal staining

SA-β-gal activity was determined using the BioVision Senescence Detection Kit (Milpitas, CA; #K320-250). For each condition and replicate, cells were counted, and 7,000 cells were seeded into 8-well culture slides coated with poly-lysine (Corning, Corning, NY; #354632). After 24 hours, the SA-β-gal staining assay was performed as per the manufacturer's protocol. For each experiment, approximately 100–150 cells were counted.

## RNA extraction and quantitative real-time PCR

Total RNA was prepared using the PureLink Micro-to-Midi total RNA Purification System (Invitrogen, Carlsbad, CA; #12183018A), according to the manufacturer's protocol. Samples were first treated with DNase I Amp Grade (Invitrogen, Carlsbad, CA; #18068015) to eliminate genomic DNA contamination. RNA was reverse transcribed into cDNA using a High-Capacity cDNA Reverse Transcription Kit (Applied Biosystems, Foster City, CA; #4368813) according to the manufacturer's protocol. Quantitative real-time PCR (qRT-PCR) reactions were performed as described using the Universal Probe Library system (Roche, Basel, Switzerland). Actin and tubulin predeveloped TaqMan assays (Applied Biosystems, Foster City, CA) were used to control for cDNA quantity. qRT-PCR assays were performed on the LightCycler 480 System (Roche, Basel, Switzerland). The primers and probes were as follows: Human actin, F 5′-CCAACCGCGAGAAGATGA, R 5′-TCCATCACGATGCCAGTG, UPL probe #64; Human tubulin, F 5′-CTTCGTCTCCGCCATCAG, R 5′-TTGCCAATCTGGACACCA, UPL Probe #58; Human IL-6, F 5′-GCCCAGCTATGAACTCCTTCT, R 5′-GAAGGCAGCAGGCAACAC, UPL Probe #45; and Human p16$^{INK4a}$, F 5′-GAGCAGCATGGAGCCTTC, R 5′-CGTAAC-TATTCGGTGCGTTG, UPL Probe #34.

## Exosome characterization and size distribution analysis

Protein determination is performed on exosomes/EVs isolated by ultracentrifugation by direct absorbance, and 20 μg of protein is used for input for the MacsPlex Exosome Kit (Miltenyi, Bergisch Gladbach, Germany) assay. These exosomes are enriched for CD63, CD9, and CD81 surface proteins using antibody beads. This pool of exosomes is then probed for 34 other surface markers used for analysis and comparison across samples. Particle diameter and concentration were assessed by tunable resistive pulse sensing (TRPS) on an IZON (Christchurch, New Zealand) qNano Nanoparticle Characterization instrument using an NP150 nanopore membrane at a 47 calibration with 110-nm carboxylated polystyrene beads at a concentration of $1.2 \times 10^{13}$ particles/mL (Zen-bio, Research Triangle, NC).

## Processing, quantification, and statistical analysis of MS data

DIA acquisitions were quantitatively processed using the proprietary Spectronaut v12 (12.020491.3.1543) software [32] from Biognosys (Schlieren, Switzerland). A panhuman spectral library was used for Spectronaut processing of the DIA data [76]. Quantitative DIA MS2 data analysis was based on extracted ion chromatograms (XICs) of 6–10 of the most abundant fragment ions in the identified spectra. Relative quantification was performed comparing different conditions (senescent versus control) to assess fold changes. To account for variation in cell number between experimental groups and biological replicates, quantitative analysis of each sample was normalized to cell number by applying a correction factor in the Spectronaut settings. Cell number and correction factors applied to each experiment are available in S6 Table. The numbers of replicates for each experiment are as follows: X-irradiated fibroblasts, 4 senescent and 4 control replicates; X-irradiated epithelial cells, 5 senescent and 5 control replicates; 4-day RAS-induction fibroblasts, 10 senescent and 10 control replicates; 7-day RAS-induced fibroblasts, 6 senescent and 6 control replicates; atazanavir-treated fibroblasts, 3 senescent (9 days of treatment), 3 senescent (14 days of treatment), and 4 control replicates; X-irradiated fibroblast exosomes, 5 senescent and 5 control replicates; and 7-day RAS-induced fibroblast exosomes, 6 senescent and 6 control replicates. Significance was assessed using FDR-corrected q-values <0.05.

## Pathway and network analysis

Gene ontology, pathway, and network analysis was performed using the GlueGO package, version 2.5.3, in Cytoscape (https://cytoscape.org/), version 3.7.1 [33,34]. Curated pathways for enrichment analysis were referenced from the following databases: GO Biological Function, GO Cellular Compartment, Kegg pathways, WikiPathways, and Reactome Pathways. For gene ontology data, testing was restricted to pathways with experimental evidence (EXP, IDA, IPI, IMP, IGI, IEP). The statistical cutoff for enriched pathways was Bonferroni-adjusted $p$-values <0.01 by right-sided hypergeometric testing. Pathway-connecting edges were drawn for kappa scores >40%. Kappa scores are a measure of inter-pathway agreement among observed proteins that indicate whether pathway agreement is greater than expected by chance based on shared proteins. Pathways with the same color indicate ≥50% similarity in terms.

## K-means clustering

Unsupervised clustering was performed in Python with Scikit-learn, a module integrating a wide range of machine learning algorithms [77]. Datasets were preprocessed with the StandardScaler function and clustered with the KMeans algorithm.

## Data visualization

Violin plots were visualized in R (https://www.r-project.org/) using the "ggplot2" package [78]. Venn diagrams were constructed using the "VennDiagram" package [79]. Color palettes in R were generated with the "RColorBrewer" package [80]. Pathway and network visualizations were generated and modified using the GlueGO package in Cytoscape [33,34].

## Supporting information

**S1 Fig. Senescence markers induced by IR, RAS, and ATV.** (A) Representative images of SA-β-Gal staining of senescent and control (quiescent) primary human lung fibroblasts and renal epithelial cells following induction of senescence by either IR, RAS, or ATV. (B) Quantification of SA-β-Gal–positive cells. (C) Levels of p16INK4a and IL-6 mRNAs determined by qPCR and expressed as fold change of senescent over control (red line) cells. (D) Commonly reported SASP factors for each inducer, cell type, and fraction. Red up-arrow = significantly increased (q-value < 0.05), blue down-arrow = significantly decreased (q-value < 0.05), hyphen (-) = not significant. ATV, atazanavir treatment; Epi, renal epithelial cell; Fib, fibroblast; IL-6, interleukin 6; IR, X-irradiation; qRT-PCR, quantitative real-time PCR; RAS, inducible RAS overexpression; SA-β-Gal, senescence-associated β-galactosidase; SASP, senescence-associated secretory phenotype.
(TIF)

**S2 Fig. Cell viability assays.** Amount of cell death over a 24-hour period as determined by Sytox Green viability dye assay or propidium iodide inclusion assay.
(TIF)

**S3 Fig. Western blot confirmation of top core SASP factors.** (A) Western blot exposures of top core SASP factors, GDF15, STC1, SERPINE1, and MMP1, in non-senescent control fibroblasts, early senescent fibroblasts (4 days of RAS induction), and fully senescent fibroblasts (7 days of RAS induction). (B) Densitometry analysis of western blot. *$p$-value < 0.05 versus CTL. CTL, quiescent control; GDF15, growth/differentiation factor 15; MMP1, matrix

metalloproteinase-1; RAS, inducible RAS overexpression; SASP, senescence-associated secretory phenotype; SERPINE1, plasminogen activator inhibitor 1; STC1, stanniocalcin 1.
(TIF)

**S4 Fig. Distribution of log2-fold changes in IR-induced fibroblast and epithelial cell secretomes.** (A) Violin plot of all significant (q-value < 0.05) log2-fold changes in the sSASP of IR-induced fibroblasts and epithelial cells. (B) Histogram of all significant log2-fold changes in the sSASP of IR-induced fibroblasts and epithelial cells. IR, X-irradiation; sSASP, soluble senescence-associated secretory phenotype.
(TIF)

**S5 Fig. Exosome/EV proteomic markers and size distribution analysis.** (A) Table of exosome- and EV-specific markers identified in exosome and soluble fractions of fibroblasts by mass spectrometry. Multiple peptides from defining exosome/EV markers were identified in the exosome fractions of RAS- and IR-induced senescence experiments, but none were detected in the soluble fractions. (B) Table showing EVs secreted per cell, average EV diameter, and standard deviation of EV diameter in senescent and control cells in complete (10% FBS) medium and low-serum (0.2% FBS) medium. The mean diameter of each condition is significantly different from each other condition (p-value < 0.00001, two-tailed t test). (C) Size distribution analysis of EVs secreted by senescent and control cells in complete and low-serum medium. (D) Exosome/EV-specific markers detected in isolated EV fractions in each treatment group, as measured by MACSPlex exosome detection kit. (E) Median levels of every surface marker measured in exosome/EV fractions by MACSPlex exosome detection kit. EV, extracellular vesicle; FBS, fetal bovine serum; IR, X-irradiation; RAS, RAS oncogene overexpression.
(TIF)

**S6 Fig. Comparison of proteomic and transcriptomic changes in the fibroblast SASP.** Transcriptomic changes in the SASP of fibroblasts reported in a recent meta-analysis [24] (Hernandez-Segura and colleagues, 2017) were compared with proteomic changes in the SASP of the current study. (A) Comparison of transcriptomic meta-analysis and proteomic analysis of secretomes in IR-induced senescent cells compared with non-senescent cells. (B) Venn diagram comparing RAS-induced senescence changes at the transcriptome and secreted proteome level. (C) Venn diagram of the core senescent transcriptome signature (genes changed at senescence regardless of inducer) versus changes common to IR- and RAS-induced senescence at the secreted proteome level. (D) Venn diagram comparing the senescent transcriptome and secreted proteome core signatures. IR, X-irradiation; RAS, RAS oncogene overexpression; SASP, senescence-associated secretory phenotype.
(TIF)

**S1 Table. Mass spectrometry quantification for each dataset as separate worksheets in a single excel workbook.**
(XLSX)

**S2 Table. Proteins with significantly increased secretion in response to all senescence inducers.**
(XLSX)

**S3 Table. Proteins with significantly increased secretion in all cell types in response to all senescence inducers.**
(XLSX)

**S4 Table. Age-associated plasma proteins also present in the SASP as determined in our proteomics experiments.** SASP, senescence-associated secretory phenotype.
(XLSX)

**S5 Table. Reagents and resources.**
(DOCX)

**S6 Table. Cell culture details for each experiment, including seeding density, culture vessel, cell counts, and correction factors.**
(XLSX)

**S7 Table. Inducer-specific secretome, transcriptome, and combined protein/RNA signatures for IR and RAS-induced senescent fibroblasts.** IR, X-irradiation; RAS, inducible RAS overexpression.
(XLSX)

**S1 Data. Underlying numerical data for each figure.**
(XLSX)

**S1 Raw Images. Raw western blot images.**
(PDF)

## Acknowledgments

We thank John C.W. Carroll for graphical support generating figures.

## Author Contributions

**Conceptualization:** Nathan Basisty, Luigi Ferrucci, Judith Campisi, Birgit Schilling.

**Data curation:** Nathan Basisty, Vagisha Sharma.

**Formal analysis:** Nathan Basisty, Chirag Rao.

**Funding acquisition:** Judith Campisi, Birgit Schilling.

**Investigation:** Nathan Basisty, Abhijit Kale, Ok Hee Jeon, Chisaka Kuehnemann, Therese Payne, Anja Holtz, Samah Shah.

**Methodology:** Nathan Basisty.

**Software:** Nathan Basisty, Vagisha Sharma.

**Supervision:** Nathan Basisty, Birgit Schilling.

**Validation:** Nathan Basisty.

**Visualization:** Nathan Basisty, Chirag Rao.

**Writing – original draft:** Nathan Basisty.

**Writing – review & editing:** Nathan Basisty, Luigi Ferrucci, Judith Campisi, Birgit Schilling.

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
