## [Editor Report · Decision Letter 0]

20 Aug 2019

Dear Dr Schilling, 

Thank you for submitting your manuscript entitled "A Proteomic Atlas of Senescence-Associated Secretomes for Aging Biomarker Development" for consideration as a Methods and Resources article by PLOS Biology.

Your manuscript has now been evaluated by the PLOS Biology editorial staff and an expert and I am writing to let you know that we would like to send your submission out for external peer review.

*Please be aware that, due to the voluntary nature of our reviewers and academic editors, manuscripts may be subject to delays during the holiday season. Thank you for your patience.*

Please re-submit your manuscript within two working days, i.e. by Aug 22 2019 11:59PM.

Kind regards,

Hashi Wijayatilake, PhD,

Managing Editor

PLOS Biology

---

## [Decision Letter · Decision Letter 1]

20 Sep 2019

Dear Dr Schilling,

Thank you very much for submitting your manuscript "A Proteomic Atlas of Senescence-Associated Secretomes for Aging Biomarker Development" for consideration as a Methods and Resources article at PLOS Biology. Your manuscript has been evaluated by the PLOS Biology editors, an Academic Editor with relevant expertise, and by several independent reviewers.

In light of the reviews (below), we will not be able to accept the current version of the manuscript, but we would welcome resubmission of a revised version that takes into account the reviewers' comments. We ask that you address Reviewer 3's comments to the best of your ability but we will not insist on new proteomic data. As an additional request from the Academic Editor, please do also clarify whether the “plasma ageing markers” referred to in Fig. 5 (published by Tanaka et al.) are up- or down-regulated in the Tanaka signature, or remove the down-regulated markers from the analysis. 

Please note that we cannot make any decision about publication until we have seen the revised manuscript and your response to the reviewers' comments. Your revised manuscript is also likely to be sent for further evaluation by the reviewers.

Your revisions should address the specific points made by each reviewer. Please submit a file detailing your responses to the editorial requests and a point-by-point response to all of the reviewers' comments that indicates the changes you have made to the manuscript. In addition to a clean copy of the manuscript, please upload a 'track-changes' version of your manuscript that specifies the edits made. This should be uploaded as a "Related" file type. You should also cite any additional relevant literature that has been published since the original submission and mention any additional citations in your response. 

Before you revise your manuscript, please review the following PLOS policy and formatting requirements checklist PDF: http://journals.plos.org/plosbiology/s/file?id=9411/plos-biology-formatting-checklist.pdf. It is helpful if you format your revision according to our requirements - should your paper subsequently be accepted, this will save time at the acceptance stage.

Please note that as a condition of publication PLOS' data policy (http://journals.plos.org/plosbiology/s/data-availability) requires that you make available all data used to draw the conclusions arrived at in your manuscript. If you have not already done so, you must include any data used in your manuscript either in appropriate repositories, within the body of the manuscript, or as supporting information (N.B. this includes any numerical values that were used to generate graphs, histograms etc.). For an example see here: http://www.plosbiology.org/article/info%3Adoi%2F10.1371%2Fjournal.pbio.1001908#s5.

For manuscripts submitted on or after 1st July 2019, we require the original, uncropped and minimally adjusted images supporting all blot and gel results reported in an article's figures or Supporting Information files. We will require these files before a manuscript can be accepted so please prepare them now, if you have not already uploaded them. Please carefully read our guidelines for how to prepare and upload this data: https://journals.plos.org/plosbiology/s/figures#loc-blot-and-gel-reporting-requirements.

We expect to receive your revised manuscript within two months. Please email us (plosbiology@plos.org) to discuss this if you have any questions or concerns, or would like to request an extension. At this stage, your manuscript remains formally under active consideration at our journal; please notify us by email if you do not wish to submit a revision and instead wish to pursue publication elsewhere, so that we may end consideration of the manuscript at PLOS Biology.

When you are ready to submit a revised version of your manuscript, please go to https://www.editorialmanager.com/pbiology/ and log in as an Author. Click the link labelled 'Submissions Needing Revision' where you will find your submission record. 

Sincerely,

Hashi Wijayatilake, PhD, 

Managing Editor

PLOS Biology

REVIEWS:

Reviewer #1: 

The paper is well written and is could be another step to identify a molecular signature for aging biomarkers. The effort to identify aging biomarkers that could be used as general markers for all aging-related conditions could be a chimera. Indeed, authors clearly evidenced many differences in the SASP contents of the senescent phenotypes they analyzed. Nevertheless, some markers may be present in many aging-related conditions and this paper gives indication about that. In my opinion, the efforts to identify common aging biomarkers should also consider that every senescent phenotype as a unique molecular profile (see for example: Oncotarget. 2015 Nov 24;6(37):39457-68. doi: 10.18632/oncotarget.6277. PMID: 26540573). I mean starting from differences could better lead to find common features. Authors should address this issue in the discussion.

--

Reviewer #2: 

PLOS Biology

Title: A proteomic Atlas of Senescence-Associated Secretomes for Aging Biomarker development. 

In this study, Basisty et al., show a comprehensive proteomic analysis on the components of the SASP using two different cell types and three different inducers of cellular senescence. After an exhaustive analysis, the authors determine that the composition of the SASP seems to be depending on the cell type as well as the type of inducer used to induce senescence. Also, the authors have found that senescent cells release more exosomes than control cells and that their protein composition differs from the secretory SASP (sSASP). Finally, by comparing their aSASP's data obtained in cells to the study using biomarkers in human plasma for diseases and aging, the authors conclude that a significant amount of SASP’s components can be considered as a good biomarker for aging and diseases.

Although this is a well written and very interesting work, I felt the authors could go a little more in deep about the meaning of their results., i.e., differences respect sSASP vs. exosomes, how they interpret the differences found in specific pathways in fibroblasts vs. epithelial cells or sSASP vs. exosomes, how the sSASP, that is intracellular phenomenon can be correlated to factors found in the plasma?

Overall, I found this work will significantly move the field forward with respect to cellular senescence and it links to biomarkers for aging and diseases. 

Minor comments 

1. It called my attention that the authors used 10% CO2, there is a specific reason for that.

2. What would be the importance of having two different compartments (cytosolic) and extracellular vesicles containing SASP components? Which one will be more relevant as a biomarker for aging and diseases?

3. Interestingly miRNA has been involved as an important cargo of exosomes, however, the authors didn't describe if miRNAs were part of the exosomes' analysis. Did they analyze this? 

4. Why the exosomes released by senescent cells induce either by IR or RAS have such a different proteomic composition? On the same experiment, is no clear whether the pathway analysis (Figures B and C) are the analysis on one type of stressor or whether it is an average between data from IR and RAS.

5. Using the K means clustering, the authors suggested 3 proteins can be used as a surrogate biomarker of sSASP (lines 218 and 219). However, one of them, CXCL1, in epithelial cells has an opposite change with what was found in fibroblasts. Therefore, this is indicating that these 3 proteins can be used as biomarkers of the SASP for human fibroblasts, and maybe only the IMR-90. Wandering whether each cell type will have their own set of proteins that will serve as a biomarker of SASP to that type of cell.

6. What would be the hypothetical mechanism that allows sSASP proteins (secreted intracellularly) be also found in plasma?

Reviewer #3: 

In this article, Basisty et al describe the development of a “proteomic atlas” of senescence-associated secretory pheotypes, which is currently online at www.saspatlas.com. The authors state that current knowledge of SASP factors is limited, due mainly to the array-based and transcriptional-based approaches used to detect soluble SASP factors. Correspondingly, they hypothesize that the SASP is more specific to source and condition than previously appreciated. Thus, the authors aim to stock their proteomic database with SASP factors derived from multiple senescence inducers and cell types, as measured using an unbiased mass spectrometry approach. They begin by inducing cellular senescence via RAS ovexpression, ATV treatment, or IR in primary human lung fibroblasts or primary human renal epithelial cells in vitro. The authors then utilize mass spectrometry to quantify corresponding proteomic changes in secreted soluble factors and exosomal content in comparison to control cells. They find that induction of senescence leads to an overall increase in levels of SASP factors independent of induction method. However, they determine that classical SASP proteins respond differently in epithelial cells compared to fibroblasts, with many classical SASP proteins unchanged or even decreased in epithelial cells. Furthermore, when comparing the SASP induced by IR, RAS, or ATV treatment in fibroblasts, the authors find only a limited number of conserved changes (termed core senescence factors) with the majority exhibiting heterogeneous changes. Finally, the authors analyze exosomal SASP, and find that IR and RAS induce distinct proteomic signatures in eSASP. 

Critique:

1. At present, the data provided by the authors does not sufficiently address the gaps in knowledge that they lay out in the introduction. Specifically, the authors state three main aims: 1) “to develop robust and specific senescence and aging biomarkers” with “a comprehensive profile of the context-dependent and heterogenous SASP” (lines 60-61), 2) to build an “in-depth, quantitative and comparative assessment of SASPs originative from multiple stimuli and different cell types” (lines 87-88), and 3) to assess the “proteomic content and function of exosomes…secreted by senescent cells” (lines 91-92). As noted below, a number of issues arise. 

• Towards aim 1, the authors state that they are generating an “Atlas of senescence-associated secretomes for aging biomarker development.” To that end, they identify a list of >1000 genes that are “associated with senescence” due to their largely heterogeneous changes induced by IR, RAS, or ATX treatment in fibroblasts or epithelial cells. They then go on to note that 101/217 aging factors identified in Tanaka et al, 2018 overlap with this list, positing a connection between senescence secretomes and aging blood biomarkers. However, given the heterogeneous and large nature of the senescence secretomes they report, we take issue with the grouping of all proteomic changes into one list, termed new SASP. Additionally, given that this paper is framed in the context of aging, it is difficult to say how meaningful the new SASP is to physiological aging, given that IR, RAS-OE, and ATX treatment are not particularly physiological drivers of aging. To appropriately make these claims or comparisons, it would be necessary, and of great interest, to add the secretome of senescent cells and non-senescent counterparts from an aged mouse or aged human to the database.

• Towards aim 2, the authors fail to fully explore context specificity of the SASP, even within the limited scope of renal epithelial cells and lung fibroblasts. Given that the epithelial cell IR secretome shows major differences compared to the fibroblast IR secretome, it would be of great interest to test how the other senescence inducers affect the epithelial secretome. 

• Towards Aim 3, the authors characterize the content of exosomes released by IR- vs RAS-induced senescent fibroblasts. Beyond listing the proteins present and comparing them, the authors do not conduct any informative analysis. They propose intriguing hypotheses that “exosome/EV proteins might distinguish senescent cells of different origins or resulting from different stressors” (lines 303-304); however, they do not assess whether any of the SASP factors might be predictive of cell source or senescent induction type. The authors should consider further analyzing their own dataset, or comparing it to published datasets (however limited by technical approach) to determine if such a stress/source signatures exist. 

2. The reasoning and details of the methodological approaches chosen by the authors are not clearly described. 

• It is unclear why the authors chose human lung fibroblasts and renal cortical epithelial cells for their studies. In the introduction, the authors mention that senescent cells are relevant to aging and disease. Are these cell types specifically relevant to those conditions? 

• For the detection of senescence-related secreted proteins, the authors profile conditioned media by mass spectrometry and use a cutoff value of q-value <0.05 and a 1.5-fold change (SEN/CTL). It is unclear if these values are somehow first normalized to cell number prior to comparing protein abundance. If not, it is impossible to determine whether the changes observed are due to altered secretions or artifacts related to differential cell abundance. This is of concern given that the clear majority of changes are observed in the same direction (upregulation). Additionally, if the images in figure S1A are representative, it appears cell density is not the same between CTL and SEN induced cells. These issues also apply to analysis of exosomal content. 

• The methods are lacking in experimental detail. For example, they do not report the density at which cells were plated. Please ensure that all information needed to reproduce the results of this study are included in the methods.

• Figure 1 suggests that senescence was induced in fibroblasts and epithelial cells by IR, RAS, ATV, prior to downstream proteomic analysis of secreted factors and exosomes. However, this is misleading as RAS overexpression or ATV treatment were only conducted on cultured fibroblasts, and not on epithelial cells. Similarly, exosomes were only collected from IR- or RAS-fibroblasts. 

3. The text and figures are frequently vague, making it difficult for the reader to draw meaningful conclusions about the data being presented. 

• The authors state that (line 174) “the magnitude of the fold-changes in the sSASP were generally higher in fibroblasts than in renal epithelial cells, regardless of inducer” (2A). It is unclear where magnitude is represented in figure 2A. If it is the bold numbers with arrows, then the numbers in figure 2A are different than those in the text. To state that the magnitudes of the fold-changes in the sSASPs significantly higher in fibroblasts than epithelial cells, it needs to be supported by statistical approaches and concrete numbers. It is further unclear how the authors reach the conclusion that this change is independent of inducer, when epithelial cells only recieved IR. The authors should make sure to clearly explain what conditions/cell types, etc. were used in their analysis, and to accurately represent their results. 

• Throughout the entirety of the text, phrases such as “most proteins” (line 125), “large fraction” (line 146), and “a more even mix” (line 173) need to be replaced with specific numbers and statistics. 

• The authors frequently round in the text (ex: “~1700”- line 145). In some cases, it represents improper rounding such as: “about 68 per cell and 49 per control cell” (line 294) rounded from 49.9. This raises concerns as to the rigor of the analysis. 

• The authors confirm the presence of beta-galactosidase, p16, and IL-6 expression in each of their conditions (figure S1A-C). They then state that “there was no detectable cell death” (line 131, figure S2). First, this is not reflected in the data: Figure S2 shows viability at ~90% and cell death at ~5%. It is also unclear what cell type and condition is being assessed in figure S2. Cell viability should be reported for all models of senescence, as cell death could be a major contributor to the secretome assessed. 

4. The combinations of data employed for pathway analyses throughout the paper are confusing. The fact that the data were combined at all is principally in conflict authors’ idea that the SASP is cell type and senescence-induction specific. Moreover, the conclusions drawn from the pathway analyses presented throughout the paper are primarily speculation, given the lack of experimental data provided to support them. 

• The authors perform “pathway and network analysis of proteins increased in the sSASPs of [IR] fibroblasts and [IR] epithelial cells” and conclude that “many pathways were commonly enriched in both the epithelial and fibroblast sSASPs (Fig 3C and Fig 2C)” and that “pathways enriched uniquely by epithelial cells included protein translation and degradation.” These conclusions suggest that the authors did pathway and network analysis on the increased sSASPs from fibroblasts and epithelial cells separately, however this is not what is represented in figure 3C or 2C. 

• The authors conclude, for example, that the “enrichment of neurodegeneration-associated proteins and pathways suggests that senescent cells contribute to neurodegenerative diseases” (lines 209-211). This conclusion is very misleading to the reader, as the authors 1) claim cell-type specificity but are working with fibroblasts and epithelial cells to draw conclusions about pathological, aged brain cells and 2) the authors conducted no experimental analysis to validate these findings, or even cross-reference their capacity to be “biomarkers” with data from published plasma biomarker studies. 

5. The database itself is an interesting idea, and the current interface is well-made. The graphics generated by the SASP query tool is intuitive and helpful. However, exporting the data is a bit difficult. Currently, downloading the data table generates an excel populated with what appears to be the logRatio – however this is not clearly labeled. It would be nice to have the downloaded table include the logRatio and -log10(qvalue).

Minor. 

• Figure 4B: The percent label is vague. What is this a percentage of?

• Figure S4: Where is CD63, CD9, etc. values for CTL fractions?

• Figure S4: The authors report CD81 and CDC42 values in figure S4, but do not discuss in the text.

• Figure S4: The authors state that the mean diameter and size distribution of senescent and control exosomes/EVs were similar. Please provide statistics to show that they are not different. 

• Figure S4: Graphs depicting the concentration and diameter of exosomes should use the same x-axis to better facilitate the comparison. 

• There are numerous grammatical errors. For example, “Several types of stress elicit a senescence and SASP response, which in turn can drive multiple phenotypes and pathologies associated with mammalian.” (line 63).

--

Reviewer #4: 

This is an interesting and valuable report. It is essentially descriptive but the quality and extent of the analyses is relevant. Particularly the finding of common factors present in the SASP and in the plasma of old donors. I have a few comments that the authors may want to address:

1. Perhaps is worth mentioning in the abstract that soluble and exosome SASPs have been analyzed.

2. Regarding this sentence in the Introduction: “We also present the first comprehensive proteomic analysis of the exosomal SASP (eSASP), which is largely distinct from the sSASP”. Please avoid claims of priority and acknowledge (cite, compare and discuss) data in https://www.ncbi.nlm.nih.gov/pmc/articles/PMC6613042/ (incidentally this study identifies 1,600 proteins in the eSASP)

3. Typos: Fig S1D is mislabeled as S1C, and it says LAMB1 instead of LMNB1.

4. The nomenclature is confusing: “secretome” or “SASP” sometimes refers exclusively to soluble SASP (sSASP) and not exosomes (for example Fig. 2A classifies “secretome” vs “exosomes”), however in other places both are considered SASP (sSASP vs eSASP). There are multiple instances in the text where the meaning has to be inferred from the context. Please, be consistent and simple. 

5. Fig. 2A: please clarify the meaning of the numbers:

a. The first and second columns, do they refer to proteins that change both in fibros and epi combined?

b. The third column (labelled “fibro” and “epi”), does it refer to proteins that change with any of the three stressors combined? The fact that the numbers in “fibro” coincide exactly with the numbers in the secretome of IR-senescent cells, is it just a coincidence? Or a typo? 

6. In line 332: “core SASP” (SASP components resulting from all senescence inducers). I would rather say: “core SASP” (SASP components common to all senescence inducers).

---

## [Decision Letter · Decision Letter 2]

2 Dec 2019

Dear Dr Schilling,

Thank you for submitting your revised Methods and Resources entitled "A Proteomic Atlas of Senescence-Associated Secretomes for Aging Biomarker Development" for publication in PLOS Biology. I have now obtained advice from the original reviewers and have discussed their comments with the Academic Editor. Based on the reviews, we will probably accept this manuscript for publication, assuming that you will correct the manuscript to address the remaining points raised by Reviewer 4 and carefully revise the entire manuscript to catch all such inconsistencies and errors. Please also make sure to address the data and other policy-related requests noted at the end of this email.

We expect to receive your revised manuscript within two weeks. Your revisions should address the specific points made by each reviewer. In addition to the remaining revisions and before we will be able to formally accept your manuscript and consider it "in press", we also need to ensure that your article conforms to our guidelines. A member of our team will be in touch shortly with a set of requests. As we can't proceed until these requirements are met, your swift response will help prevent delays to publication.

*Copyediting*

*Published Peer Review History*

*Early Version*

*Submitting Your Revision*

Sincerely,

Hashi Wijayatilake, PhD, 

Managing Editor

PLOS Biology

REVIEWS:

Reviewer #1: 

None

Reviewer #2 (Viviana I Perez): 

In this revised manuscript the authors addressed all previous reviewers’ questions and concerns, which resulted in an strengthen manuscript.

Reviewer #3: 

For the most part, the authors have satisfactorily addressed the majority of our concerns.

Reviewer #4: 

Authors have satisfactorily addressed my comments.

However, authors have to revised carefully their data because there are still errors. I have not revised everything, but the few things I checked have errors. A few examples regarding Fig. S1D and the corresponding Table S1.1:

- LMNB1 in AVT-SEN-fibros: the figure says it goes up, however the table says that it goes down

- LAMB1 on the contrary goes up in all three conditions. I wonder if the authors wrongly select as UP in all conditions the protein LMNB1, when it should have been LAMB1

- HMGB1 in exosome of IR-SEN-fibros: the figure says it goes up, the table says it goes down

- HMGB1, MMP2, TIMP1/2, IGFBP3/4/5/7: the figure says all go down, the table says all go up

Please, revise everything. I can understand a few errors, but these seem to me as too many.

---

## [Editor Report · Decision Letter 3]

13 Dec 2019

Dear Dr Schilling,

On behalf of my colleagues and the Academic Editor, Manuel Serrano, I am pleased to inform you that we will be delighted to publish your Methods and Resources in PLOS Biology. 

PRESS 

Kind regards,

Hannah Harwood

Publication Assistant, 

PLOS Biology

on behalf of

Hashi Wijayatilake,

Managing Editor

PLOS Biology